# Insights from human NF-κB knockouts

Maximilian Pfisterer [ID][1,3], Jan Dreute [ID][1,3] & M Lienhard Schmitz [ID][1,2]✉

## Abstract

The well-studied NF-κB signaling system is a key mediator of the inflammatory response. Large-scale sequencing studies in humans now allow initial insights into non-essential human genes in which both alleles carry mutations that prevent protein expression or function. Here, we compiled the non-essential genes identified in various sequencing studies and analyzed the occurrence of knockouts in the human NF-κB signaling system. This revealed a lower knockout frequency in the NF-κB system compared to the entire genome. Since drugs inhibiting NF-κB pathway components were unsuccessful in clinical trials so far, the naturally occurring knockouts of NF-κB and its upstream regulators could provide new candidates for therapeutic intervention. To investigate the potential functional importance of posttranslational modifications (PTMs) occurring on NF-κB components, we analyzed not only their evolutionary conservation but also, as a second criterion, their genetic constraint in the sequenced individuals. This approach revealed the absence of missense mutations at key modification sites involved in NF-κB activation and identified additional candidate sites for future studies.

**Keywords** NF-κB; Gene Essentiality; Knockout; Loss-of-function; Signaling
**Subject Categories** Genetics, Gene Therapy & Genetic Disease; Signal Transduction

## Introduction

Cellular stress triggers the activation of specific transcription factors that mediate adaptive responses to maintain cellular homeostasis. Stress-induced transcription factors include nuclear factor erythroid 2-related factor 2 (NRF2, activated by oxidative stress), hypoxia-inducible factor (HIF), p53 (activated by DNA damage), and the NF-κB family, which plays an important role in the induction of the immune response (Hayden and Ghosh, 2012; Oren and Prives, 2024; Ortmann et al, 2024; Tonelli et al, 2018). The core NF-κB system consists of five different DNA-binding subunits (p65, RelB, c-Rel, p105, and p100) and six proteins of the inhibitor of NF-κB (IκB) family, namely IκBα, IκBβ, IκBε, IκBζ, Bcl3, and IκBNS. In addition, the core NF-κB system includes the activation machinery consisting of two kinases (IκB kinase (IKK)α, IKKβ) and the scaffold protein NF-κB essential modulator (NEMO/ IKKγ). The NF-κB system is already found in ancient multicellular organisms such as sponges and cnidarians and serves to protect the integrity of multicellular organisms (Hayden and Ghosh, 2008; Williams and Gilmore, 2020). In addition to its role in innate and adaptive immunity, NF-κB protects cells against damaging and noxious agents, thus typically antagonizing the induction of cell death. Discovered almost 40 years ago by Ranjan Sen and David Baltimore (Sen and Baltimore, 1986), NF-κB studies have regularly contributed to the discovery of general principles in signal transduction.

The NF-κB activation pathway is based on two fundamental processes. One pathway proceeds by proteolytic degradation of inhibitory IκB proteins, which retain NF-κB DNA-binding subunits in the cytosol. The second pathway leads to the production of the DNA-binding subunits p50 and p52 through their proteolytic release from the precursor proteins p105 and p100. Both activation processes start with inducible phosphorylations at specific sites, leading to the formation of phospho-degrons. The phosphorylated regions then serve as docking sites for ubiquitin E3 ligases, which leads to degradation of the IκB proteins (Alkalay et al, 1995; Chen et al, 1995) or allows the processing of the precursor proteins (Heissmeyer et al, 1999; Xiao et al, 2001). In both cases, the NF-κB subunits are transported to the nucleus where they bind their respective target sequences and trigger gene expression.

Further fundamental studies revealed the existence of various feedback loops, which were deciphered experimentally and characterized through computational modeling (Basak et al, 2012; Wang et al, 2022). Additional key mechanisms include the induction of signaling cascades by supramolecular complexes or filaments and signal transduction via protein–protein complexes that can be transiently formed through PTMs, such as the formation of regulatory ubiquitin chains that serve as binding platforms (Chen, 2005; Lin et al, 2010).

This wealth of biochemical information is complemented by the availability of a large and likely unparalleled number of sophisticated mouse models exhibiting loss-of-function (LOF) and gain-of-function (GOF) alleles, which display a wide range of features, most of them associated with inflammatory conditions and their direct and indirect consequences (Pasparakis et al, 2006). It is therefore not surprising that the cell-intrinsic and extrinsic functions of NF-κB have been associated with numerous human diseases, including a broad spectrum of inflammatory diseases and cancers. The role of NF-κB in tumors is very complex, as on one

[1]Institute of Biochemistry, Justus-Liebig-University, Giessen, Germany. [2]Member of the CPI and the German Center for Lung Research, Giessen, Germany. [3]These authors contributed equally: Maximilian Pfisterer, Jan Dreute. ✉E-mail: lienhard.schmitz@uni-giessen.de

hand it can support an anti-tumor microenvironment and assist T cells and NK cells in tumor defense, while on the other hand its cell-intrinsic activity protects tumor cells from apoptosis (Riedlinger et al, 2018; Taniguchi and Karin, 2018). While mutations in the core NF-κB system rarely function as driver mutations in cancer (Perkins, 2007), deregulation of NF-κB can cause a variety of different, largely inflammation-associated effects (Online Mendelian Inheritance in Man, OMIM). Disease-causing NF-κB mutations in humans typically result in hypomorphic changes (causing decreased expression and/or activity), resulting in LOF of varying severity (Schnappauf and Aksentijevich, 2020). Although it is plausible that many of these mutations partially impair functionality of NF-κB, the molecular mechanisms are often not well defined or understood (Schnappauf and Aksentijevich, 2020; Zhang et al, 2017). Beyond the descriptive identification of NF-κB mutants, more information is needed on residual activities, intracellular localization and changes in the interactome and functions of the altered protein. A complete inactivation was only described for c-Rel where splice-site mutations lead to the generation of severely truncated mutants that could not be detected by immunoblotting (Beaussant-Cohen et al, 2019), while hypermorphic alleles were described for IκBα (Courtois et al, 2003).

While these genetic alterations were discovered through sequencing of diseased patients, deep sequencing of >1 million healthy individuals now provides completely new insights into the redundancy of the NF-κB signaling system in human physiology. These studies allow the identification of tolerable gene losses in the NF-κB core system as well as at the level of its upstream regulators.

## Gene essentiality

At the level of single-celled organisms, the concept of gene essentiality refers to the ability to proliferate, while in the case of multicellular organisms this definition refers to genes that are indispensable for reproductive success and maintenance of a normal phenotype (Bartha et al, 2018; Rancati et al, 2018). Overall, essential genes are more conserved between species, have fewer paralogues, are intolerant to variation, and their gene products are more central in protein–protein interaction networks (Costanzo et al, 2016; Davierwala et al, 2005).

In *Saccharomyces cerevisiae*, individual deletion of all genes showed that only ~20% of the genes are required for viability (Giaever and Nislow, 2014). The essentialomes of additional species were determined using systematic perturbation of gene function by RNAi, insertional mutagenesis, or CRISPR-Cas. In multicellular organisms, the percentage of essential genes ranges from ~25% in *Caenorhabditis elegans* (Kamath et al, 2003; Qin et al, 2018) to ~37% in *Drosophila melanogaster* (Port et al, 2020; Spradling et al, 1999). These numbers compare well to mice, where approximately one-third of all genes are essential for life (Ayadi et al, 2012; Dickinson et al, 2016; Tian et al, 2018). All these studies have defined gene essentiality under laboratory conditions in the absence of challenges such as food scarcity, predator stress, infections or diseases. Therefore, it would not be surprising that the number of essential genes is somewhat higher under real-life conditions, but at least half of the genes appear to be dispensable.

## Identification of non-essential genes in humans

Since the genes necessary for the health and fertility of an adult human cannot be determined experimentally by targeted gene inactivation, the identification of non-essential genes in humans has been advanced through large sequencing studies, which are summarized in Table 1.

These sequencing results provide natural models for the inactivation of human genes. Here, we focus on gene essentiality in adult humans and do not consider knockout genes identified through genetic screens in tumor cell lines, as these reflect a disease state rather than the genes required for the functioning of an entire organism representing human physiology. The first studies in humans focused on selected groups through the sequencing of either bottlenecked populations that emerged from small founding groups (Lim et al, 2014; Sulem et al, 2015) or consanguineous populations with high degrees of parental relatedness (Narasimhan et al, 2016a; Saleheen et al, 2017). These datasets were recently extended by sequencing a large number of individuals from outbred and diverse populations (Karczewski et al, 2020; Lek et al, 2016; Sun et al, 2024), now allowing annotation of biallelic predicted LOF (pLOF) mutations including indels, frameshift, nonsense, or canonical splice-site mutations. These mutations are expected to result in LOF, either due to the complete absence of protein expression or the presence of significant truncations that are predicted to severely compromise protein function. These data, despite the limitations and caveats given in Box 1, provide exciting initial insights into the tolerability of gene losses in adult humans. Here, we only consider mutations affecting both alleles of a gene as either homozygous or compound heterozygous mutations, where both alleles harbor different mutations. Our analysis focuses on protein-coding genes and excludes non-coding RNAs, as they have many and often ill-defined targets.

## Functional consequences of human gene knockouts

The results of these sequencing studies suggest that a healthy average human carries a total of approximately 100 pLOF mutations with 20-50 genes completely inactivated by homozygous or compound heterozygous mutations (Lek et al, 2016; Narasimhan et al, 2016b; Saleheen et al, 2017; Sun et al, 2024). Approximately half of all mutations were observed as singletons, i.e., they occur only in one individual (Lek et al, 2016; Sun et al, 2024). Additional information about the health status of the sequenced individuals is highly variable. In many studies, the health status of the individuals examined was not recorded, and potential health problems may only appear later in life. In the ExAC study, individuals with severe pediatric diseases were excluded (Lek et al, 2016). In another study, LOF carriers showed no changes in drug prescription rates or clinical staff consultation rates (Narasimhan et al, 2016a), which is considered as a good proxy for the health status. Only the UK Biobank study combines genetic analysis with deep medical phenotyping (Bycroft et al, 2018).

Generally, the functional consequences of pLOF alleles span a wide spectrum. Some gene knockouts can be linked to diseases (Bycroft et al, 2018), some of which emerge only later in life or after

**Table 1.** Overview of large human sequencing studies considered for this work.

| Reference, acronym | Sequenced individuals | # Knockout genes | Health status of knockouts |
|---|---|---|---|
| MacArthur et al, 2012 | 185 whole genomes | 98 | Not investigated |
| Lim et al, 2014 | 3000 exomes, bottleneck population | 80 | Not investigated |
| Sulem et al, 2015, deCODE Iceland | 104,220 whole genomes, bottleneck population | 1171 | Not investigated, 2 of 38 children of heterozygous parents were recorded to die in their first year. |
| Lek et al, 2016, ExAC The knockout genes were identified by re-analysis (Narasimhan et al, 2016a) | 60,706 exomes | 1775 | Severe pediatric diseases were excluded. |
| Narasimhan et al, 2016a, ELGH | 3222 exomes, consanguineous population | 781 | Knockout carriers have no changes in drug prescription rate and clinical staff consultation rates, for some participants, health data are available. |
| Saleheen et al, 2017, Promis | 7078 exomes, consanguineous population | 1317 | Few individuals with mutations were further analyzed. |
| Karczewski et al, 2020, gnomAD | 125,748 exomes and 15,708 genomes | 1815 | Not investigated |
| Rausell et al, 2020 | Re-analysis of ExAC and gnomAD databases | 166 | Not investigated |
| Sun et al, 2024, RGC-ME, and reprocessed data from UK Biobank (Backman et al, 2021) and Mexico City Prospective Study (Ziyatdinov et al, 2023) | 983,578 exomes | 4848 | Not investigated |

The acronyms, as well as information on the number of sequenced individuals, the identified knockout genes, and the availability of health status data are provided. Lists of knockout genes are provided in the cited studies, while the knockout genes in the ExAC dataset (Lek et al, 2016) were identified through re-analysis by Narasimhan and colleagues (Narasimhan et al, 2016a).

the onset of disease-causing events. For example, interferon regulatory factor 7 (IRF7)-deficient individuals are healthy, but highly susceptible to life-threatening influenza infections (Ciancanelli et al, 2015). However, LOF mutations can also have positive consequences, as exemplified by the chemokine receptor type 5 (CCR5) gene, which confers protection against infection with the human immunodeficiency virus (Liu et al, 1996). A further example is the occurrence of LOF mutations in Caspase-12, which increase resistance to sepsis (Xue et al, 2006). The occurrence of most homozygous pLOF genes in humans is without any recognizable functional consequence due to the inherent redundancy of genes, which contributes to robustness and tolerance to exogenous disturbances (Hunter, 2022). In such cases, isoforms, related proteins and alternative pathways can fully compensate for the function of the lost protein.

We leveraged the data from the different studies listed in Table 1 and identified 6689 non-redundant human knockout genes, from which 3895 were only identified in one dataset, including 2736 genes identified in the "Regeneron Genetics Center Million Exome (RGC-ME)" project (Sun et al, 2024). The other 2794 genes were identified in 2 or more datasets (Fig. 1A,B; Appendix Fig. S1). All knockout genes are given in Dataset EV1. The pLOF genes represent one-third of the estimated 20,000 human protein-coding genes, thus making up a significant portion of the non-essential human genes.

The compiled set of non-essential genes was subjected to an over-representation analysis of functional annotations using GO (gene ontology) and Reactome databases. The analysis for GO class "biological process" showed enrichment of genes related to olfactory and metabolic pathways affecting the turnover of fatty acids and other catabolic pathways (Appendix Fig. S2). These broad categories are also overrepresented in the non-essential genomes of other organisms (Bartha et al, 2018). Despite the limitations discussed in Box 1, the dataset generated by this compilation is large enough to perform specific analyses of individual signaling pathways such as NF-κB.

## Human knockouts in the core NF-κB pathway

To investigate human mutations in the NF-κB system, we categorized it into the core system consisting of the DNA-binding subunits, the IκBs and the IKK complex as well as the upstream systems composed of the various regulators mediating NF-κB activation in several distinct activation pathways (Dataset EV2). We plotted the gene essentiality score $S_{het}$ (selection coefficient against heterozygous LOF) (Sun et al, 2024) for the NF-κB core group and its upstream regulators (given in Dataset EV3) in comparison to 16,710 annotated genes. As controls, we also determined gene essentiality for gene groups with high (splicing machinery) or low (olfactory) essentiality. This revealed an elevated essentiality of genes encoding core NF-κB components and a slightly higher essentiality of the different upstream NF-κB systems in comparison to the global genome (Fig. 2).

Since genes of the NF-κB system neither belong to the group of essential genes (the essentialome) nor to the strongly redundant genes, such as the olfactory receptors, we examined the individual knockouts more closely. Within the core system, biallelic knockouts were predicted for p65 (Lek et al, 2016; Narasimhan et al, 2016a), IκBε (Saleheen et al, 2017), IκBζ and IKKα (Sun et al, 2024).

**Box 1  Caveats**

The datasets contain errors in annotations, databases and DNA sequencing results. In addition, differences in coverage, along with the inability to span breakpoints, reduce the sensitivity for calling of larger deletions.

There is some overlap between the datasets; the RGC-ME project (Sun et al, 2024) re-analyzed data from other studies, as summarized in Table 1.

A given mutation might not lead to a complete LOF, especially when non-conserved exons are affected and truncated proteins are predicted. In these cases, there is no information on the stability, localization, interactome, or potential residual activity of the truncated protein.

The various studies sequencing exomes or whole genomes differ in details of the employed assumptions, scores and mathematical metrics (Bartha et al, 2018; Rancati et al, 2018). Key metrics used to determine genes essentiality are pLI (probability of being LOF intolerant), LOEUF (LOF observed/expected upper bound fraction) and $S_{het}$, as described in detail elsewhere (Zeng et al, 2024).

Essentiality of genes can also depend on differences in epigenetic or genetic or backgrounds, therefore, the deletion of a gene can have distinct effects in different individuals. For example, a gene can become essential due to the loss of a second gene (synthetic lethality). Conversely, the essentiality of one gene can depend on a second gene, for example, when one gene protects against the toxic effects of the second gene (Chen et al, 2016; Nijman, 2011). As humans carry several homozygous pLOFs, secondary effects of knockout combinations are inherent to the data.

The low allele frequencies of the observed (point) mutations often do not allow for a robust statistical analysis and therefore have the character of a case report.

The occurrence of genetic mosaicism cannot be excluded. Together, these considerations led to the proposal that gene essentiality should not be considered a binary and static property, but rather a dynamic and fluidly definable one (Bartha et al, 2018; Rancati et al, 2018).

While deletion or mutation of IκBε has not yet been described, the other three knockouts in the core NF-κB system have been linked to diseases. Truncations of the C-terminal transactivation domains of p65 are known to cause Behçet's disease, a rare, chronic, and multi-system inflammatory disorder (Adeeb et al, 2021; Badran et al, 2017; Lecerf et al, 2022). It would be interesting to know whether the 21 persons with a p65 knockout (due to a biallelic stop downstream of Asp347) identified in the study by Lek and colleagues also suffer from an inflammatory disease. In addition, mutations in IκBζ have been associated with ulcerative colitis and psoriasis (Coto-Segura et al, 2017; Kakiuchi et al, 2020; Nanki et al, 2020), while mutations in IKKα lead to defects in the development of skin epidermis (Lahtela et al, 2010). Further information on the clinical relevance of mutations in NF-κB knockouts are given in Dataset EV4.

How is it possible that knockout carriers of these NF-κB pathway genes are presumably symptom-free, while mutations in these genes are associated with inflammatory diseases? (I) The type of genetic alteration has a decisive impact on the resulting phenotype. A well-studied example is the knockout of the gene encoding the cellular prion protein (PrP), which shows no obvious phenotype in mice (Büeler et al, 1992). In contrast, mice with a point mutation in the PrP protein show spontaneous PrP aggregation and develop Spongiform encephalopathy (Sigurdson et al, 2011). The functional difference between the complete loss of a protein and its mutated form is also relevant to human disease. An example of this is the

p53-coding gene, which is rarely deleted in tumor patients but in the vast majority of cases exhibits oncogenic p53 missense mutations that enable the gain of new harmful functions, thereby promoting cancer development (Lane, 2024). Many further studies observed incomplete matching between human gene knockouts and disease-causing gene mutations. For example, it is well known that deafness can be caused by *GJB2* gene mutations (omim.org (Wilcox et al, 2000)), while individuals with a *GJB2* knockout exhibit normal audiometry (Narasimhan et al, 2016a). For further examples describing a lack of congruence between disease-causing mutations and human knockouts, we refer to the relevant literature (Narasimhan et al, 2016b).

(II) Another explanation is derived from the observation that the impact of a specific gene loss also depends on its interaction with the genome. The penetrance of a given mutation is often dependent on the genetic context, where so-called modifier genes, which do not cause disease on their own, can influence the effect of a disease-causing mutation. A well-studied example is cystic fibrosis transmembrane conductance regulator (*CFTR*) gene mutations, which can cause cystic fibrosis (CF) depending on the levels of transforming growth factor (TGF)-β1 (Najm et al, 2024).

## Limited comparability of human and murine gene knockouts

To compare non-essential genes between humans and mice, we utilized our compiled list of 6689 non-essential human genes alongside data from the International Mouse Phenotyping Consortium, which classifies knockouts of 8565 genes as viable, subviable, or lethal (Dickinson et al, 2016; Tian et al, 2018). We included only genes with viable knockouts, reducing the dataset to 5641 genes. Our analysis specifically focused on murine genes with single orthologs in the human genome, as such orthologs are more likely to retain conserved biological functions and enable more reliable cross-species comparisons. This filtering refined the final mouse dataset to 4913 and the human dataset to 4723 non-essential genes. Notably, only 1870 (38%) of these genes were shared between the two species (Appendix Fig. S3). The considerable size of the discordant sets reflects not only the differences between synthetic experiments and data derived from outbred human populations, but also fundamental differences between these organisms.

The significant differences in knockout tolerability between humans and mice are illustrated by several examples. While the *Lrig3* gene is essential for the formation of the lateral semicircular canal in mice (Abraira et al, 2008) and the *Otop1* gene is crucial for the formation of murine otoliths (Hurle et al, 2003), both genes can be inactivated in humans without any apparent effects (Sulem et al, 2015). The same is true for the histone methyltransferase PRDM9 which leads to meiotic defects and thus infertility in mice (Brick et al, 2012; Hayashi et al, 2005), while in humans an individual identified with LOF variants in the *PRDM9* gene was healthy and fertile (Narasimhan et al, 2016a). These – admittedly handpicked – examples align well with the general picture describing limitations in the comparability between human and murine models (Seok et al, 2013). Accordingly, the results of drug tests in preclinical animal models are poor predictors of human responses (Ingber, 2020; Pound and Ritskes-Hoitinga, 2018). Up to 90% of drugs that are successful in mice

**A**

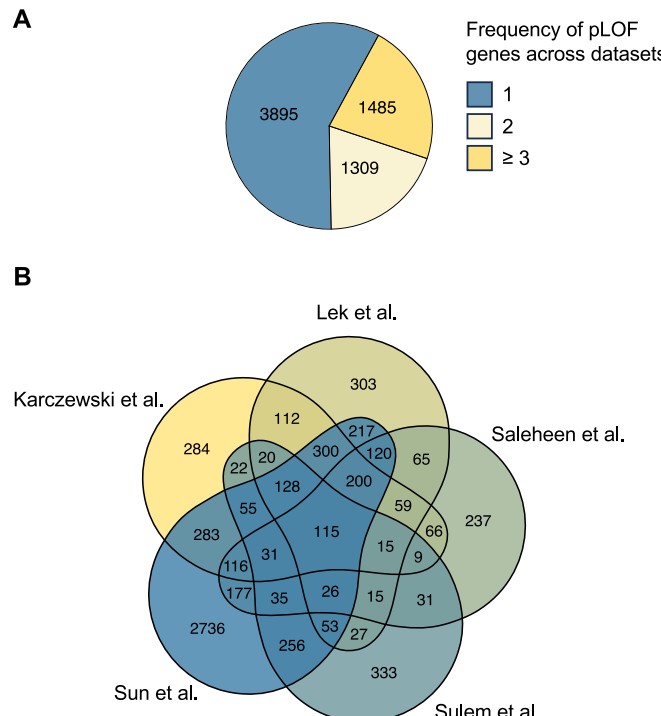

**B**

Figure 1.  **Summary of knockouts identified in the different sequencing studies.**

(A) Knockout genes identified in the sequencing studies listed in Table 1 were compared with each other. The number of knockout genes identified in only one study (blue), two studies (light yellow), or three or more studies (yellow) is shown. (B) Venn diagram displaying the overlap of knockout genes in the five largest sequencing studies.

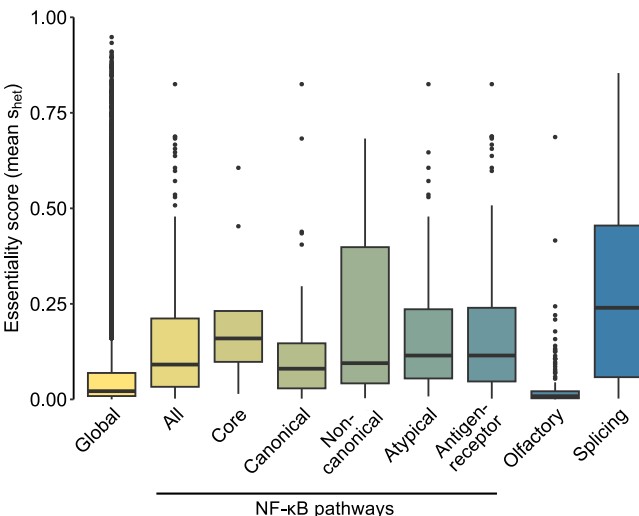

Figure 2.  **Essentiality scores of the core and upstream NF-κB pathways.**

Based on the $S_{het}$ value as modeled by Cassa et al (2017), we utilized $S_{het}$ annotations from Sun et al (2024) to assess gene-specific constraints. Specifically, we mapped $S_{het}$ scores to individual genes from defined gene lists (Dataset EV2) and to gene sets from MSigDB (R version 7.5.1), including Splicing (GO:0048024) and Olfactory Signaling (hsa04740). The analysis shows the subgroups forming the NF-κB core system and the various activation pathways, as well as the overall group derived from them, referred to as "All".

do not have the desired efficacy and safety in humans. Conversely, well-established medications in humans, such as Aspirin, exert their pharmacological effect in mice only with exceedingly high doses and would never have passed preliminary tests in mice (Chen et al, 2018; de Jong and Maina, 2010). Without devaluing the many important insights gained from mice, these findings also urge for a deeper understanding of signaling systems such as NF-κB in human physiology.

## New NF-κB-targeting drugs through human knockouts?

The biochemical and genetic elucidation of various signaling pathways over the past decades has, in some cases, allowed the successful development of highly specific and FDA-approved drugs. Examples include the phosphoinositide 3-kinase (PI3K) inhibitor Idelalisib, which is used for the treatment of relapsed chronic lymphocytic leukemia (CLL) (Cheah and Fowler, 2016), and the Janus kinase inhibitor Tofacitinib for the treatment of inflammatory diseases such as rheumatoid arthritis (Sandborn et al, 2012). Interestingly, despite significant efforts from basic researchers and several pharmaceutical companies, no inhibitor with specificity for the NF-κB pathway has yet found its way into clinical use. All clinically used drugs with effects on NF-κB (such as

Bortezomib – proteasome inhibition; Corticosteroids – activation of steroid hormone receptors) exert their effects only indirectly as a side effect of the directly targeted structure. The failure to develop NF-κB-specific drugs may be attributed to the fundamental importance of this signaling pathway for human physiology. Interestingly, also non-essential genes can serve as drug target structures, as an earlier study showed that approximately half of 383 FDA-approved drugs are directed to the products of unconstrained genes (Minikel et al, 2020).

An interesting example of the inspiring and hypothesis-generating effects of knockouts in drug development is provided by the *PCSK9* gene, where nonsense variants are associated with low levels of low-density lipoprotein (Cohen et al, 2005). This led to the development of the two PCSK9-targeting and FDA-approved drugs Alirocumab and Evolocumab which are currently used for the treatment of hypercholesterolemia (McDonagh et al, 2016).

## Human knockouts in the different NF-κB activation pathways

With the exception of a few cell systems, including B cells, hair follicles and neurons, NF-κB is kept in an inactive state by association with inhibitory IκB proteins (Henkel et al, 1993; Kanarek and Ben-Neriah, 2012). Specific stimuli can trigger NF-κB activity with peculiar kinetics and amplitudes, typically followed by the induction of negative feedback loops to shut down the response (Renner and Schmitz, 2009). Inappropriate termination or persistent increased basal activity can contribute to ailments and is observed in many tumor cell lines or in parainflammation

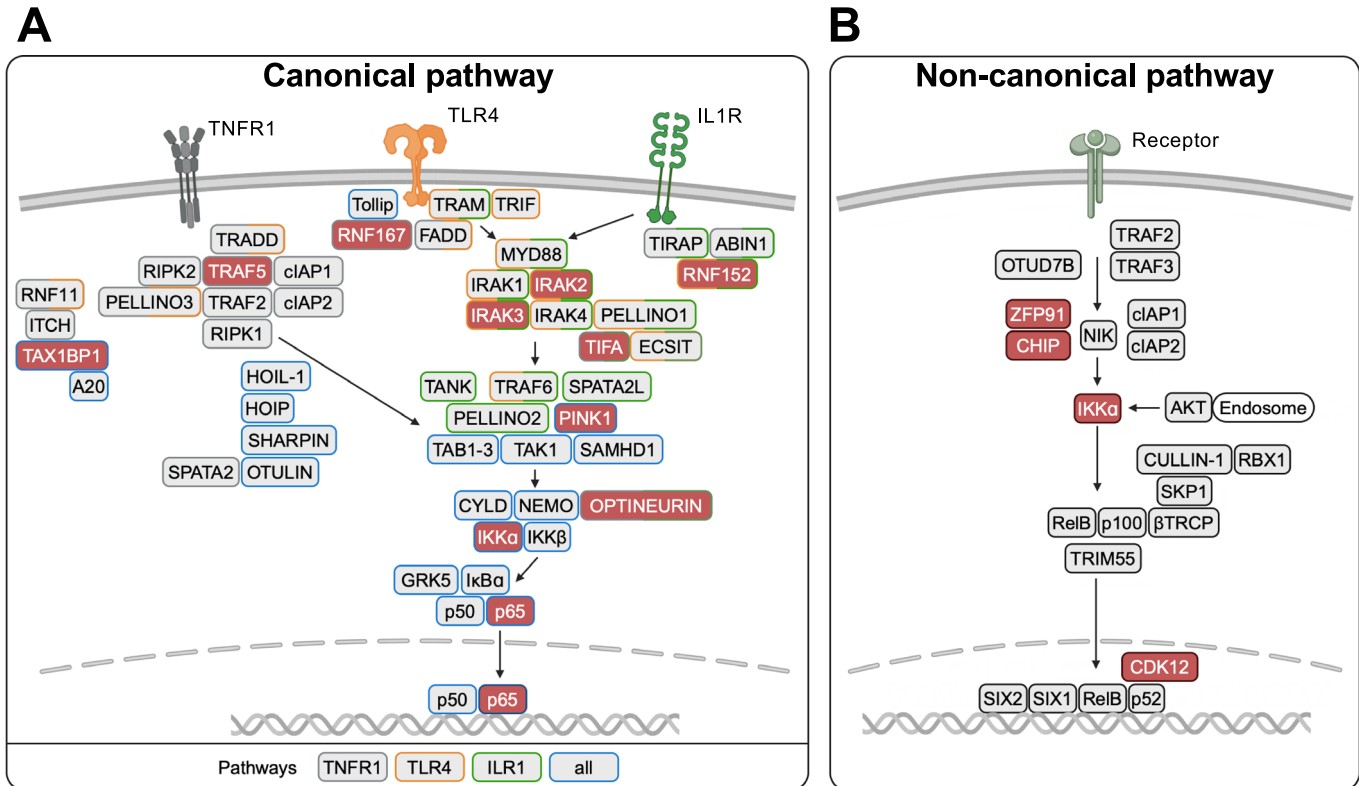

**Figure 3. Schematic representation of regulatory proteins contributing to the up- or downregulation of different NF-κB pathways.**

Here we only considered proteinous regulators of the NF-κB pathways in healthy cells. Excluded from the analysis were: The membrane receptors with their directly associated cofactors, as they also feed into other pathways. Factors that were identified only based on pharmacological inhibitors or through large screens without confirmatory follow-up experiments. Cofactors of (post)transcriptional processes, as they lack of specificity for NF-κB. The schematic representation of the complexes was created based on literature data compiled in Dataset EV2, which also provides the references and respective gene names. In most cases, the stoichiometry and details of the protein–protein interactions are not known. The genes encoding the proteins highlighted in red have been described as human knockout genes. The canonical (**A**) and non-canonical (**B**) NF-κB pathways are shown.

(Medzhitov, 2008). Like most signaling pathways, the components of NF-κB-regulating pathways show a certain degree of redundancy by distributing functions across networks to confer robustness, buffering and resilience (Uda et al, 2013). Here, we focus on a few well-characterized NF-κB activation pathways triggered by specific events in healthy and unmutated cells (Fig. 3).

(a) The canonical NF-κB activation pathway is induced by different immune receptors, as summarized in excellent reviews (Hayden and Ghosh, 2008; Sun and Ley, 2008) and schematically shown in Fig. 3A. In innate immunity, recognition of PAMPs (pathogen-associated molecular patterns) or DAMPs (damage-associated molecular patterns) by specific receptors triggers a signaling cascade leading to the activation of NF-κB. The early response induced by LPS (lipopolysaccharide)-activated TLR4 (Toll-like receptor 4) signaling proceeds via the MyD88 (myeloid differentiation primary response 88)-dependent pathway. MyD88 then recruits IRAK1 (Interleukin-1 receptor-associated kinase 1) and IRAK4 (Interleukin-1 receptor-associated kinase 4) which, together with other proteins, assemble a multifunctional supramolecular organizing center termed Myddosome (Motshwene et al, 2009). This clustering leads to the activation of the ubiquitin E3 ligase TRAF6 (TNF receptor-associated factor 6), which attaches Lys63-linked ubiquitin chains to its interaction partners. These

ubiquitin chains function like a molecular glue for transient interactions, as they interact with specific domains contained in interaction partners, including members of the TAB (TAK1-binding protein) family. TAB adapter proteins directly bind the kinase TAK1 (TGFß-activated kinase 1) and facilitate its auto-activation, ultimately leading to the activation of IKKs, a process dependent on the addition of linear polyubiquitin chains to the NEMO scaffold protein by the linear ubiquitin chain assembly complex (LUBAC) complex (Rahighi et al, 2009; Tokunaga et al, 2009). IKK-mediated phosphorylation of IκBα is then the prerequisite for its ubiquitination and proteasomal degradation. The cascade of these events ultimately results in the release of NF-κB dimers which can then translocate to the nucleus, bind to their cognate DNA and induce gene expression. Many NF-κB target genes help manage inflammatory conditions and include cytokines such as tumor necrosis factor α (TNFα) and interleukin-1β (IL-1β), which mediate and amplify inflammatory responses (Schmitz et al, 2011). The released cytokines act in autocrine and paracrine manners and bind to their cognate receptors in order to enhance and specify the inflammatory response.

Both TNFα and IL-1β receptors activate NF-κB via different signaling pathways, but share the final TAK1-mediated downstream pathway for NF-κB activation. The binding of TNFα to its

cognate tumor necrosis factor receptor 1 (TNFR1) receptor triggers its trimerization and allows recruitment of receptor-interacting protein kinase 1 (RIPK1) and TRAF2, which form a complex together with additional proteins at the cytoplasmic side of the receptor (Hayden and Ghosh, 2014). The E3 ligase function of TRAF2 mediates Lys63-linked ubiquitination of RIPK1, which leads to the induction of the TAB/TAK1 system and ultimately of the IKK complex. Also, signal transduction emanating from the IL-1-bound interleukin-1 receptor type 1 (IL-1R1) leads to the binding of MyD88, which in turn recruits IRAK1 and IRAK4. IRAK1 then activates TRAF6 which exerts its ubiquitinating function to activate TAK1 and the subsequent steps leading to liberation of the NF-κB subunits from IκB proteins (Chung et al, 2007).

The comparison between the group of non-essential genes (Dataset EV1) and activators of the canonical NF-κB signaling pathway (Dataset EV2) reveals a set of non-essential genes including IRAK2 and IRAK3 (Fig. 3A). It is plausible that the loss of a protein from a multi-protein family could be compensated by the function of another protein. Accordingly, IRAK1 and 2 were shown to have redundant roles with both individually being dispensable for LPS-induced cytokine production (Pereira et al, 2022). The non-essential proteins comprise some drug candidates which are currently in preclinical evaluation, including the IRAK2 modulator HG-1053 and the IRAK3 degrading PROTAC 23 (AstraZeneca).

(b) The non-canonical NF-κB activation pathway is less complex and only comprises a limited number of components (Fig. 3B). This pathway is activated by specific receptors of the TNF receptor superfamily including the LTβR (lymphotoxin beta receptor), CD40 (cluster of differentiation 40) and BAFF-R (B cell activating factor receptor) and is of special relevance for B cells (Senftleben et al, 2001; Xiao et al, 2001). This pathway is characterized by a delayed activation typically peaking around 12–24 h after stimulation and a sustained response. Non-essential genes encoding regulators of this pathway encode the ubiquitin E3 ligase CHIP (C-terminal Hsp70-interacting protein), the zinc finger proteins zinc finger protein 91 (ZFP91) and tripartite motif-containing protein 55 (TRIM55) as well as the kinase cyclin-dependent kinase 12 (CDK12) and IKKα (Lin et al, 2023).

(c) Antigen receptor activation of NF-κB is a critical process in adaptive immunity. The engagement of antigen receptors such as the B cell receptor (BCR) or T cell receptor (TCR) together with their respective co-receptors activates NF-κB to regulate lymphocyte proliferation, differentiation, and survival (Paul and Schaefer, 2013; Sasaki and Iwai, 2016). The involved signaling pathways employ distinct and common signal transducers (Fig. 4A). Engagement of the TCR triggers phosphorylation of its cluster of differentiation 3 (CD3) chain, allowing the recruitment of the ZAP-70 (Zeta-chain-associated protein kinase 70) kinase, which phosphorylates the adapter proteins LAT (linker for activation of T cells) and SLP-76 (SH2 domain-containing leukocyte protein of 76 kDa). This activates several downstream signaling proteins, including phospholipase C γ1 (PLCγ1), which hydrolyzes PIP2 (phosphatidylinositol 4,5-bisphosphate) into inositol trisphosphate (IP3) and diacylglycerol (DAG), an activator of protein kinase C θ (PKCθ). The activated kinase phosphorylates the so-called CBM (CARMA-BCL10-MALT1) complex, consisting of Caspase recruitment domain-containing membrane-associated guanylate kinase 1 (CARMA1), BCL10 (B cell lymphoma/leukemia) and the

paracaspase mucosa-associated lymphoid tissue lymphoma translocation gene 1 (MALT1) (Gaide et al, 2002; Oeckinghaus and Ghosh, 2009). The next step in this cascade is the TRAF6-mediated activation of TAK1 and the subsequent activation of the IKK complex in analogy to the process discussed above for the canonical NF-κB pathway. The CBM complex is also relevant for BCR-mediated NF-κB activation, with differences occurring in the receptor-proximal events. Binding of an antigen to the BCR triggers the phosphorylation of the intracellular tails of the receptor-associated Igα/Igβ heterodimer subunits by tyrosine kinases of the Src family, including B lymphoid tyrosine kinase (BLK), LYN, and FYN. This phosphorylation allows recruitment and activation of several adapter proteins and kinases, leading to activation of PLCγ2, which hydrolyzes PIP2 into IP3 and the PKCβ activator diacylglycerol (DAG) (Saijo et al, 2002). Its kinase activity then leads to activation of the CBM and downstream NF-κB. In these pathways, a number of kinases are encoded by the non-essential genome (Fig. 4A). Inhibitors for RIPK1 and BLK are still at the stage of preclinical studies, while the safety, pharmacokinetics and pharmacodynamics of the selective PKCθ inhibitor EXS4318 are currently being tested in clinical trials for patients with inflammatory diseases (Kohal et al, 2024; Yu et al, 2023).

(d) The pathways mediating NF-κB activation by DNA damage differ significantly from other pathways, as the initial signals originate from damaged DNA in the nucleus, undergo signal processing in the cytosol, and ultimately return to the nucleus (McCool and Miyamoto, 2012; Wang et al, 2017), as illustrated in Fig. 4B. The damaged DNA is bound by poly(ADP-ribose) polymerase 1 (PARP1), which is activated to catalyze ADP ribosylation on itself and further substrate proteins. This allows formation of a multi-protein complex containing NEMO, protein inhibitor of activated STATy (PIASy), and the kinase ataxia telangiectasia mutated (ATM) (Stilmann et al, 2009) to trigger a signaling cascade that ultimately results in NF-κB activation. The target genes encode many anti-apoptotic genes that protect cells from death. As this function counteracts the intended induction of cell death during chemotherapy of tumors, pathway-specific inhibitors would be of great practical interest as adjuvants in oncology. Pathway-specific knockouts were detected for the genes encoding the ubiquitin-specific peptidase ubiquitin-specific protease 10 (USP10) and the kinases ATM, CLK protein kinase 2 (CLK2), and PKCη. The anti-tumor activity of CLK2 inhibitors is currently tested in cell culture models (Hu et al, 2024; Mucka et al, 2023), while ATM inhibitors such as AZD0156 entered clinical trials (STUDY ID: D6500C00001, NCT ID: NCT02588105).

In summary, the 28 knockout genes in the NF-κB system are evenly distributed across the functional groups of ubiquitination regulators (6 genes), adapters/scaffold proteins (10 genes) and protein kinases (9 genes). While kinases are frequently used drug targets with 33 kinase inhibitors and monoclonal antibodies approved by the FDA (food and drug administration) since 2018 (Li et al, 2024), recent advances in the development of proteolysis-targeting chimeras (PROTACs) now enable targeting of additional protein classes (Hsia et al, 2024).

# Regulation of NF-κB by posttranslational modifications

Historically, the NF-κB field has been groundbreaking for our understanding of the importance of PTMs in signaling pathways.

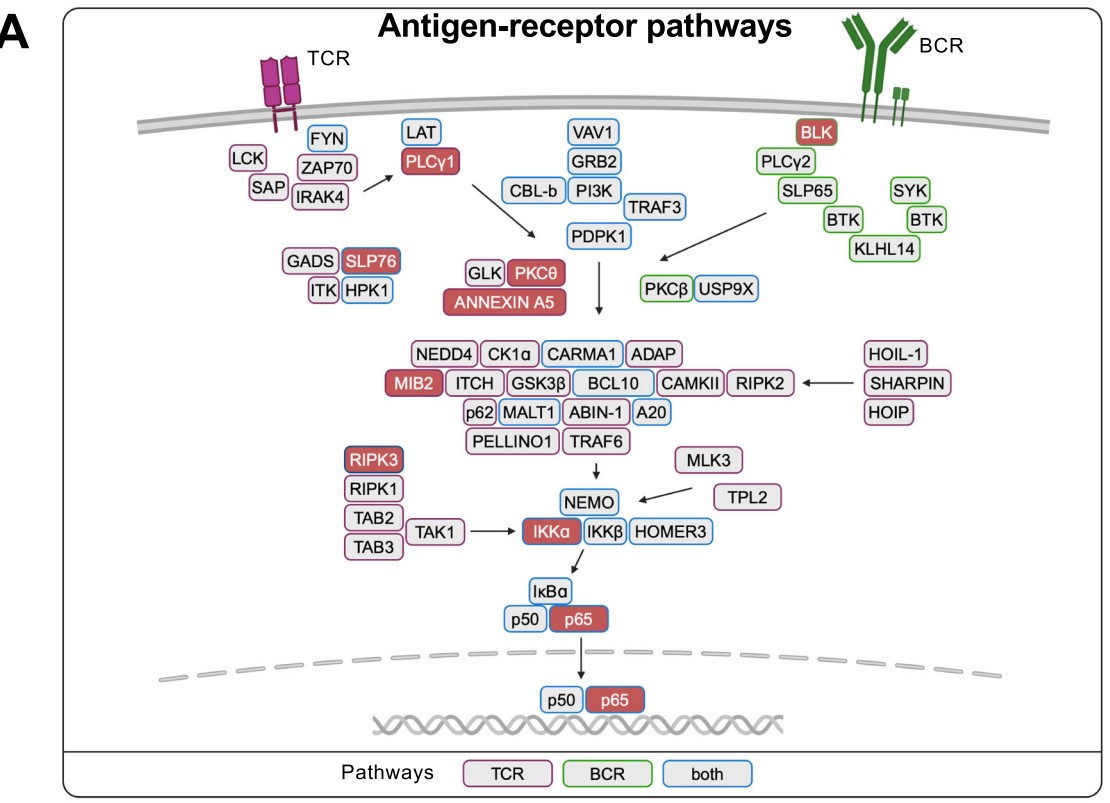

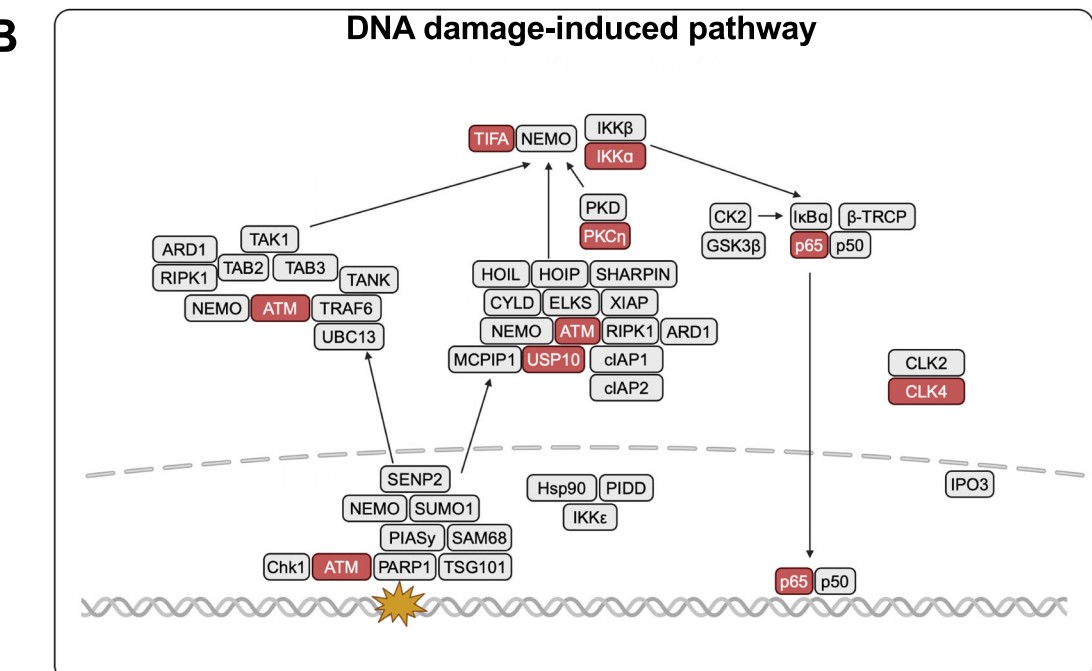

**Figure 4. Schematic representation of regulatory proteins contributing to the up- or downregulation of different NF-κB pathways.**

NF-κB activation pathways triggered by antigen receptors (**A**) and DNA damage (**B**) are shown. The criteria for NF-κB-regulating proteins correspond to those in Fig. 3, the proteins highlighted in red have been described as human knockout genes.

One fundamental NF-κB activation mechanism employs phosphorylation-coupled ubiquitination. The phosphorylation of IκBα at Ser32 and 36 is a prerequisite for the subsequent modification through ubiquitination and the degradation of the modified inhibitor by the proteasome (Brown et al, 1995; Chen et al, 1995; Scherer et al, 1995). Therefore, mutation of these key phosphorylation sites to non-phosphorylatable amino acids creates a super-repressor variant that prevents NF-κB induction in cell culture experiments and in mice (Brown et al, 1995; van Hogerlinden et al, 1999). The same regulatory principle applies to IκBβ, where phosphorylation at Ser19 and 23 is followed by ubiquitination to enable degradation of the modified protein (DiDonato et al, 1996; Weil et al, 1997).

A second fundamental NF-κB activation mechanism employs processing of the precursor proteins p105 or p100 to their DNA-binding forms p50 and p52, respectively (Heissmeyer et al, 1999; Xiao et al, 2001). Similar to IκB degradation described above, this processing is enabled by an initial phosphorylation event occurring on a patch of serine residues in the C-terminus of p105 including Ser927 and 932 (Heissmeyer et al, 2001; Heissmeyer et al, 1999; Salmeron et al, 2001). Also, inducible phosphorylation of p100 at Ser866 and 870 is the prerequisite for NIK-induced processing of the precursor protein to yield the p52 subunit (Xiao et al, 2004). Accordingly, mutation of these phosphorylation sites in p105 and p100 abrogates signal-induced processing and/or degradation (Heissmeyer et al, 1999; Xiao et al, 2004). While the importance of these initial modifications for the induction of the NF-κB response is well-studied and understood, rapid advances in mass spectrometry led to an exponential increase in the number of identified modification sites. Hundreds of PTMs occur on the components of the NF-κB core system, but their functional relevance is largely unknown. Since a number of PTMs are attached by non-enzymatic mechanisms and the majority of PTMs has no apparent function when analyzed individually (Beltrao et al, 2013; Landry et al, 2009) it is a daunting task to identify modifications with a physiological role.

## Constraint analysis of human NF-κB PTM sites

To investigate the potential functional relevance of PTM sites, we employed a two-step process by analyzing both intra-species and also the inter-species conservation, since functionally important sites are under negative selection (Josephs et al, 2015). First, we leveraged the recent availability of deep sequencing data to examine their intra-species genetic constraint, which reflects their capacity to tolerate mutations (Samocha et al, 2014). We conducted this analysis for the core NF-κB proteins and downloaded the post-translationally modified amino acids from the PhosphoSitePlus database. These were then compared with variant data from the gnomAD database to determine the frequency of missense mutations occurring at PTM sites in the sequenced individuals. This analysis identified several PTM sites that were unaffected by missense mutations in the sequenced individuals, while the observed missense mutations were typically monoallelic. This heterozygosity can be expected, as a heterozygous mutation occurring, for example, in 1‰ of individuals would result in

homozygosity in only 1 out of 4 million individuals under Hardy-Weinberg equilibrium (Gao et al, 2015).

Monoallelic PTM mutations of functionally important sites can indeed have biological effects. Since approximately 15% of human genes cannot tolerate the loss of one allele (Bartha et al, 2018), the mutation of a functionally relevant PTM site in one allele of such a gene would already lead to LOF. In addition, haploid mutations in PTM sites can exert dominant-negative effects, as exemplified by the histone H3 variant H3.3. Monoallelic mutation of Lys27 to Met in only one allele leads to changes in the modification of the remaining wild-type allele and alters gene expression, resulting in increased tumorigenesis (Chan et al, 2013). In the NF-κB system, a heterozygous missense mutation at serine 32 of IκBα has a dominant function and creates a super-repressor which causes anhidrotic ectodermal dysplasia and T cell immunodeficiency (Courtois et al, 2003).

In addition to the analysis of intra-species conservation in humans, the PTM sites not affected by missense mutations were also examined for inter-species conservation by comparing them with lemurs. These animals are a suitable model system as they share a common primate ancestor with humans and are evolutionarily positioned between humans and mice (Ezran et al, 2017). Interestingly, most PTM sites that show no missense mutations in the human sequencing data from gnomAD are conserved between lemurs and humans. These data are summarized for the NF-κB core proteins in Dataset EV5 and are exemplarily depicted for the two essential IκB proteins (Fig. 5).

Most PTM sites with known functional relevance of both IκB proteins are not found in mutated form and are also conserved in lemurs. IκBα has no alterations in both N-terminal phospho-degron sites as well as the ubiquitination site at Lys67 (Zhu et al, 2023) and of Tyr42, which is key for alternative NF-κB activation pathways (Schoonbroodt et al, 2000). Also, in the IκBβ protein, the degradation-critical amino acids Ser19 and 23 remain unmutated, just like the phospho-degrons present in the precursor proteins p100 and p105. Although these point mutations often exhibit low allele frequencies, the apparent genetic constraint of key amino acids in the human population is remarkable. Therefore, we propose that the in silico prediction of the functional relevance of PTM sites should not only consider their evolutionary conservation as proposed before (Beltrao et al, 2013), but also assess the genetic constraint within the human population.

## Conclusions and perspectives

Even though NF-κB is one of the most extensively studied signaling pathways, new experimental techniques and approaches continue to provide novel insights. Despite many open questions (see Box 2), the recent availability of large sequencing datasets encompassing more than one million individuals now provides the first sneak peek insights into non-essential human genes, including components of the NF-κB system. Characterizing the compensatory mechanisms in individuals with pLOF of NF-κB components will contribute to a deeper understanding of this signaling pathway. Data from additional sequencing studies and improved mathematical algorithms will expand the human knockout database and increase its accuracy. Relevant human mutations

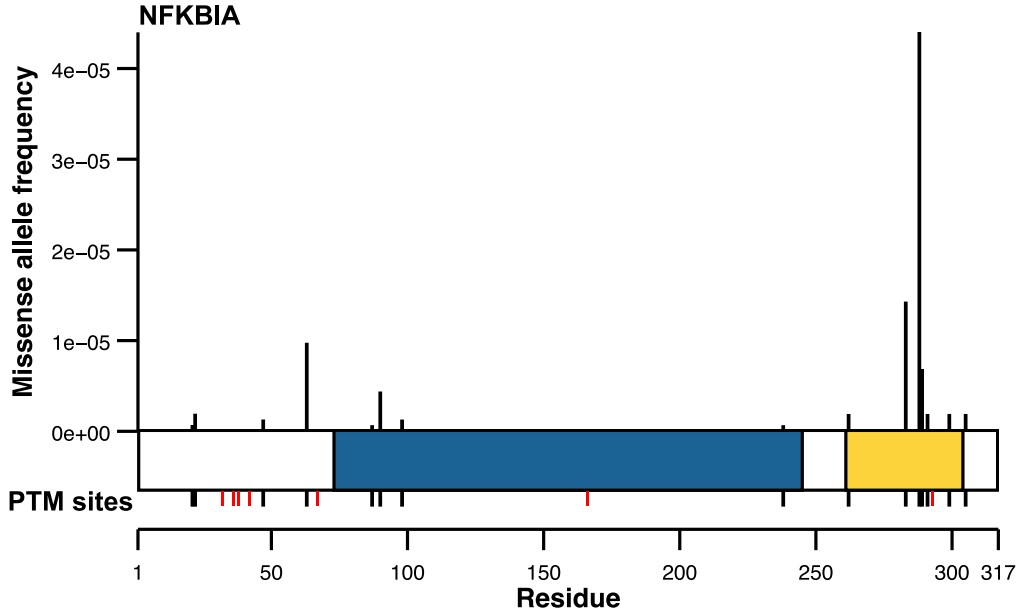

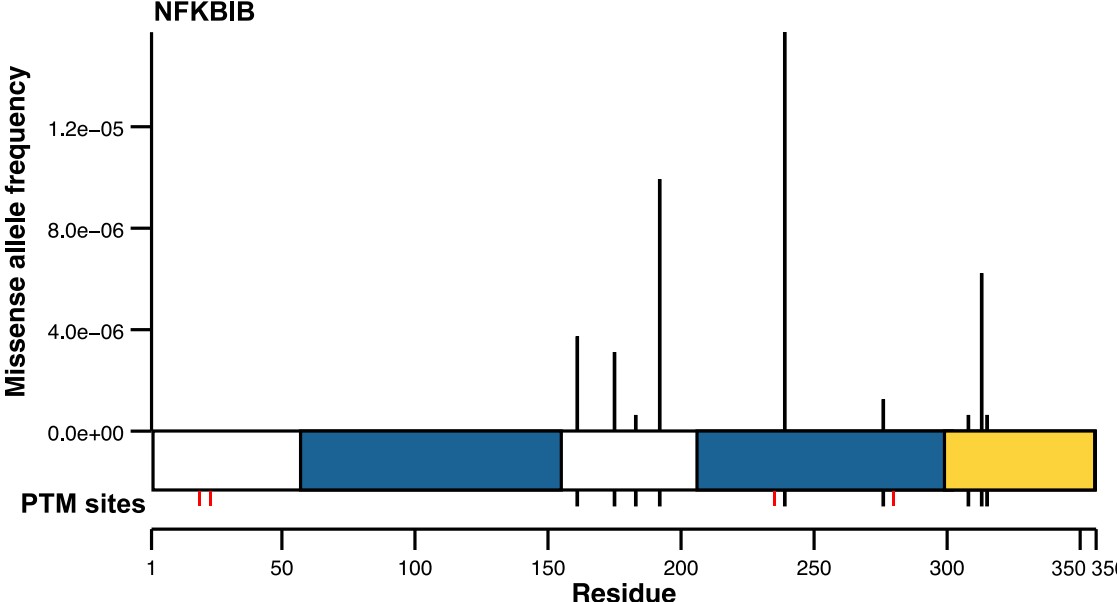

**Figure 5. Analysis of the genetic constraint for the amino acids modified by PTMs in the NF-κB core system.**

The PTMs occurring in the NF-κB core system were downloaded from the PhosphoSitePlus database and analyzed for the occurrence of mutations as well as their cumulative allele frequency (as a measure for the general mutation rate) from the gnomAD database. The code to generate the figures is available on Zenodo and can be used for any protein. The IκBα and IκBβ proteins are shown as examples, the ankyrin-repeat region is shown in blue, the PEST domain in yellow. The upper part illustrates the frequency of missense mutations, while the post-translationally modified amino acids listed in the PhosphoSitePlus database are indicated by short lines below. Amino acids under constraint (i.e., not altered by missense mutations) and conserved in lemurs were identified and are displayed as red lines. These analyses were performed for all NF-κB proteins and are given in Dataset EV5.

can be recreated and modeled in suitable model systems such as organoids or organ chips using CRISPR-Cas-mediated gene editing. This also enables further studies and facilitates the identification of genetic interactions and compensatory variants. Increasingly powerful bioinformatic tools and AI-driven models can potentially identify the yet-unrecognized role of NF-κB in

other diseases or chronic processes such as parainflammation. All these efforts could also enable the prediction and experimental validation of interventional tools to control NF-κB activity. Moreover, the genome data discussed here will also provide valuable information for the in-depth analysis of additional signaling pathways and biological processes.

> **Box 2  In need of answers**
>
> 1. Knockout of some genes might be only tolerated in the presence of naturally occurring compensatory or modifying variants. The identification of such genetic interactions and co-dependencies will also expand our understanding of the function of genes, including the NF-κB system.
> 2. To capture the spectrum of possible effects of pLOF mutants, a longitudinal follow-up of the health status and detailed phenotyping of individuals would be of great interest. Valuable insights could be generated in compliance with regulations on anonymization and data protection.
> 3. The field would benefit from the implementation of unified consensus criteria for data processing and the algorithms defining human knockouts to improve the comparability of existing and future studies.
> 4. Most studies discussed here investigated the occurrence of mutations by exome sequencing, thus covering ~1.5% of the human genome (Lander, 2011). As more than 5% of the human genome is under purifying selection and shows evidence of evolutionary constraint (Mouse Genome Sequencing et al, 2002; Sullivan et al, 2023) it can be expected that also the non-coding genome will harbor LOF variants affecting gene function. Mutation of structural or regulatory DNA elements such as enhancers and promoters can be causative for deregulated expression of individual genes or even co-regulated gene clusters (Epstein, 2009; Weiterer et al, 2020; Zaugg et al, 2022). Therefore, exome sequencing needs to be complemented by deep sequencing approaches of whole genomes to enable insights into LOF events in the non-coding genome.
> 5. Under the broad assumption that, similar to mice, more than half of the genes in humans are non-essential, many of these genes remain undiscovered. As many LOF mutations are only observed as singletons, sequencing of yet more people is required and also currently pursued by different consortia, including the European "1+ Million Genomes" initiative.

# Data availability

The code used in data analysis and figure generation is available under https://doi.org/10.5281/zenodo.15212566.

# Peer review information

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

## Acknowledgements

Our work is supported by funding from the German Research Foundation (DFG) GRK 2573/2 (RP10, Project 416910386), the Excellence Cluster CPI (Project 390649896), the EU (COST Action CA23119 Senescence2030) and the Forschungscampus Mittelhessen (2025_1_01). Figures were created with BioRender.

## Author contributions

**Maximilian Pfisterer**: Resources; Software; Visualization; Writing—review and editing. **Jan Dreute**: Resources; Software; Writing—review and editing. **M Lienhard Schmitz**: Conceptualization; Formal analysis; Funding acquisition; Writing—original draft.

## Funding

## Disclosure and competing interests statement

The authors declare no competing interests.

