## [Peer Review File · EMBO Reports]

Insights from human NF- κ B knockouts

M. Lienhard Schmitz, Maximilian Pfisterer, and Jan Dreute

Corresponding author(s): M. Lienhard Schmitz (lienhard.schmitz@biochemie.med.uni-giessen.de)

Review Timeline:

Submission Date:	4th Mar 25
Editorial Decision:	1st Apr 25
Revision Received:	15th Apr 25
Additional Reports on Revised Manuscript:	15th May 25
Authors' Response to the Additional Reports:	19th May 25
Accepted:	4th Jun 25

Editor: Achim Breiling

Transaction Report:

Dear Prof. Schmitz,

Thank you for the submission of your review article to our editorial offices. I have now received the full set of referee reports that is copied below. As you will see, all three referees state that your manuscript is interesting and timely. They have, however, several suggestions to improve the submission that I kindly ask you to address in a revised manuscript.

Given the constructive referee comments, I would thus like to invite you to revise your manuscript with the understanding that all referee points will be addressed in the revised manuscript and in a detailed point-by-point response.

Acceptance of your manuscript will depend on a positive outcome of a second round of review. It is EMBO reports policy to allow a single round of revision only and acceptance of the manuscript will therefore depend on the completeness of your responses included in the next, final version of the manuscript.

I further have these editorial requests:

- Please order the sections like this:

Title page - Abstract - Keywords - Introduction - Main text - Acknowledgements (including the funding information) - Disclosure and Competing Interests Statement - References - Table - Figure legends - Boxes

- We updated our journal's competing interests policy in January 2022 and request authors to consider both actual and perceived competing interests. Please review the policy <https://www.embopress.org/competing-interests> and update your competing interests if necessary. Please name this section 'Disclosure and Competing Interests Statement' and put it after the acknowledgment section.

- Please remove the list of abbreviations from the manuscript. Please define each abbreviation the first time it is mentioned in the text.

- Please move the Table after the references and add a legend.

- Please also note our reference format:

- We usually ask our authors to include a box called "In need of answers" that briefly outlines the major questions that are still open in a given field in the form of a few bullet points. These questions can be accompanied by a brief explanation of what would be needed to address them and may provide helpful towards setting the stage for future experimentation in the field. For an example see this recent review we published: <https://www.embopress.org/doi/full/10.1038/s44319-024-00135-4>

- Please also add callouts for the box to the manuscript text (Box 1). Please name the other Box 'Box 2' and update the callouts.

- Please move the table to the end of the manuscript text file, before Box 1.

- Please supply the supplementary material as a single pdf file labeled Appendix. The Appendix should have page numbers and needs to include a table of content on the first page (with page numbers) and legends for all content. Please follow the nomenclature Appendix Figure Sx, Appendix Table Sx etc. throughout the text, and also label the figures and tables according to this nomenclature.

- The three source data files provided are datasets. Please name these Dataset EVx, add a legend on the first TAB and upload these as dataset files. Please also add callouts to these to the manuscript text.

I think this is a very interesting review and while I appreciate that incorporating the referees' suggestions will still require some work, I am convinced that the article is worth it and will benefit from it.

When submitting your revised manuscript, we will require a Microsoft Word file (.doc) of the revised manuscript text including detailed figure legends (placed after the references), tables, but without the figures.

Please provide the final figures as separate, high-resolution files (without their legends) as .pdf, .eps, .tif, or .jpg (one file per figure). Please finalize the drafts provided and make sure they accurately illustrate the key scientific concepts that you wish to show.

Please also note the following points:

- If there are certain aspects of your figure draft that are based upon assumptions or where the scientific data remains ambiguous (for example, schematically depicting a presumed direct protein-protein interaction, protein shape or subcellular localizations etc.) please add a comment so that we can work with you on an accurate depiction. Please ensure the directionality and nature of interactions is presented accurately.
- If the figure or single panels of the figure have been adapted from a published figure, please add this information to the figure legend (e.g., 'Adapted from...' or 'Based on...'). The editor will discuss if a reference and permission will be necessary
- Please only re-use figures or parts of a figure if this is essential for understanding the concept communicated. Often a reference to a previous paper will suffice. If the figure contains re-used images or elements of images, including schematics, micrographs or photos, please make sure that you have the permission/license to publish it (this also applies to your own previous work, if the journal you published in retains copyright. Certain 'creative commons' open access licenses, such as CC-BY 4.0, allow re-use without additional formal permissions). All re-used material must be explicitly cited.
- If you use an image data base for scientific iconography (e.g., BioRender), please let us know if you have a license that allows for publication in an academic journal. Often authors use misleading iconography for expedience. Please ensure the information shown is scientifically accurate. If in doubt, please discuss with the editor or provide a sketch so that our designers can create accurate iconography. Please acknowledge the use of BioRender once in the Acknowledgements section (not in the figure legend).
- For figures created using a software for editing vector objects like Inkscape, CorelDraw etc., please send the file as a PDF (or SVG, or EPS), PowerPoint or Keynote in which the labels and objects are still editable. For figures created using Adobe Illustrator, please send the Illustrator (.ai) file.

I look forward to seeing a revised version of your manuscript when it is ready. Please let me know if you have questions or comments regarding the revision.

Yours sincerely,

Referee #1:

The manuscript by Pfisterer and colleagues leverages large genomic sequencing datasets to explore the essential or non-essential nature of genes encoding components of the NF- κ B pathway. This is a timely study given the available data from these genomic datasets and our understanding of individual components that control transcriptional responses to a diverse range of stimuli. Timely as well given the emerging realisation that mice are not human and the expanding list of genes for which there are substantial phenotypic differences between mice humans when mutated or deleted. The authors have presented a thorough, robust but very readable overview of human knockouts and the essential nature of NF- κ B components. By providing their analysis as supplementary files they also provide a resource to be utilised by others. The diagrams and figures are useful addition to what is a very interesting and thought provoking analysis. Given the relevance of NF- κ B to human disease and health this paper will be high interest to a large number of researchers in these areas. I wholeheartedly recommend it for publication.

Minor comment.

It might be helpful to explain in a little more detail how gene essentiality was calculated (shet), perhaps in the "Box" section?

Referee #2:

Pfisterer and colleagues hypothesise that non-essential genes might be good therapeutic targets because their inhibition is less likely to cause side effects. To identify such genes, they present an analysis aggregated datasets of human variants to estimate the burden of predicted LOF variants in genes relevant to NF- κ B in comparison to the remainder of the genome. The analysis is of interest but I have some reservations as outlined below:

1. In its current form, the manuscript is quite hard work. The supplementary tables are not labelled as such but appear as source files with no indication of which corresponds to each supplementary table in the text.

2. Nomenclature is at times ambiguous. For example, table 1 (and the relevant text) lists 6689 "non-redundant" genes. Two paragraphs later, the text refers to "strongly redundant genes", but these two uses of the term "(non)-redundant genes" are completely different. The former refers to the approach of compiling the gene list, and the latter refers to their biological function.
3. The relation between gene names and protein names in the NF- κ B system is inherently confusing. The authors amplify this problem because throughout the text and figures they refer to genes using their protein names. The reader is left to reconcile the information in the text and figures with the supplementary tables that consist entirely of gene names.
4. On page 14, after describing each of the NF- κ B pathways, the authors refer to the 28 knockout genes in the NF- κ B system. It is possible that this list of 28 genes is somewhere in the tables but I couldn't locate it. Here and elsewhere, the authors could consider including summary tables within the main article that identify the genes of particular interest.
5. There is some overlap between the datasets analysed (e.g. Regeneron and gnomAD). This will not affect the main findings of the analysis but will affect the interpretation of the extent of variants shared between them and should be mentioned.
6. The variants mentioned in the discussion of the core NF- κ B pathway are not all in the same category. The syndromes attributed to LoF of RELA and CHUK (Behcet's and fetal encasement/Bartsocas-Papas, respectively) are due to ultra-rare damaging variants, whereas psoriasis is only associated with polymorphisms in NFKB1. Furthermore, the analysis hinges on predictions of LOF from the aggregated datasets as indicators of their non-essentiality. While the authors acknowledge the limitations to this assumption, the analysis might be more balanced if it were to include data that has identified variants that are definitely pathogenic (e.g. ClinVar).
7. The final section on PTMs seems like a separate study. The conclusion that genetic conservation might predict functional importance is hardly novel - it is after all the basis of almost all *in silico* predictors of damage conferred by missense mutations.
8. The authors suggest that non-essential NF- κ B genes might be satisfactory therapeutic targets because their inhibition is predicted to cause fewer side effects. Another way of looking at it, however, is that inhibition of these genes might lack any therapeutic efficacy due to redundancy. The authors might comment on this possibility.

Referee #3:

Pfisterer et al. decided to write a review on NF- κ B signaling molecules as drug targets in light of the recent surge of reports on large-scale human genome sequence data. These reports show that many (nearly one third) of the nearly 18,000 protein-coding genes are completely (homozygous) or partially (heterozygous) deleted through the generation of protein truncation variants (PTVs). These truncations often result in loss-of-function (LoF) variants, which are equivalent to human 'gene knockout'. Since these LoF variants typically do not result in strong disease phenotypes, they are tolerable and non-essential.

The NF- κ B family of transcription factors plays important roles in many physiological processes. Both their inactivation and untimely activation are associated with numerous inflammatory diseases and cancer. However, efforts to intervene in these diseases by targeting NF- κ B activation pathways have not met with success. Pfisterer and colleagues wonder whether the essentiality of the NF- κ B transcription factors and many of the factors in the activation pathways might be responsible for this failure. Therefore, they focus on factors in the activation pathway that are non-essential using the curated human sequence data sets and suggest that nonessential factors are better drug targets. While the intention behind their argument is commendable, there are several fundamental flaws with the conclusions drawn by the authors. This could be a powerful review, but as it stands now, there are several flaws, including some fundamental ones, which should be carefully rectified.

Major Concerns:

First, The Genome Aggregation Database (GnomAD) Consortium published four articles in Nature in 2020. In one of these articles, by Minikel et al (doi.org/10.1038/s41586-020-2267-z), explained how human KO studies should interpret the LoF variants in drug development. Figure 1a in this article shows that the targets of 383 FDA-approved drugs contain both constrained and unconstrained genes in nearly similar proportions, as they exist in the total genome. The authors write in their summary: "Here we report three key findings regarding the assessment of candidate drug targets using human loss-of-function variants. First, even essential genes, in which loss-of-function variants are not tolerated, can be highly successful as targets of inhibitory drugs." This suggests that, as of now, there is no evidence that non-essentiality of a gene should be a criterion for a drug target. While it is possible that in the case of NF- κ B activity control non-essential factors will not be a better target, the authors must establish the logic coherently and with supporting evidence. I am totally surprised that Pfisterer et al. ignored this most relevant reference in their work. This is a significant oversight.

Second, I also do not understand the assertion that genes within the NF- κ B system are nonessential or redundant. The authors have not adequately established the criterion for redundancy. All members (I κ B, NF- κ B or IKK) in a family within the NF- κ B system show partial redundancy. To establish essentiality is even more complex. Pfisterer et al write, "biallelic knockouts were

predicted for p65", but I don't see evidence for this. Contrarily, Lek et al showed RelA has no LoF variants (Table 13). In that list, Rel, RelB, NF- κ B1 and NF- κ B2 all show LoF variants, as do IKBKB (IKK2) and CHUK (IKK1), but not RelA. Moreover, Lek et al., rarely discussed biallelic KO. Sun et al compiled a list of biallelic knockouts of 4848 genes and none of the NF- κ B family members, including RelA, appear in that list. Am I missing something here?

Additionally, the authors write that "whether the 21 persons with a p65 knockdown (due to a biallelic stop downstream of Asp347) identified by the study by Lek and colleagues also suffer from inflammatory disease" (page 9, top). I haven't analyzed their data, but the idea of having 11 biallelic LoF variants for a single gene in only 60,000 individuals seems highly improbable. The authors should check the data very carefully, considering that the 60,000 individuals are largely normal, not suffering from severe diseases. In contrast, inflammatory disorders reported by Adeeb et al. and Badran et al. are RelA heterozygous variants, which were found due to the severity of these diseases. They should double check this information.

Other comments:

The section on human knockouts in different NF- κ B activation pathways describes the pathways well but offers very little analysis of human knockouts. The 28 knockouts should be listed in a table, showing the specific mutations, whether they are truly biallelic or monoallelic, and other relevant details.

The same critique applies to the discussion of post-translational modifications (PTMs): the analysis is quite superficial. There is nothing new except the observation that many critical, highly conserved modification sites do not appear as missense variants.

Faculty of Medicine

Institute of Biochemistry, Friedrichstrasse 24, 35392 Giessen
Dr. Achim Breiling
Senior Editor
EMBO Reports

Prof. Dr. M. Lienhard Schmitz
Friedrichstrasse 24
35392 Giessen
Tel: +49 641 99-47570/1
Fax: +49 641 99-47589
lienhard.schmitz@biochemie.med.uni-
giessen.de
(<https://www.uni-giessen.de/en/faculties/f11/departments/biochemistry>)

18.04.2025

Re.: Revision of EMBOR-2024-60349V1

Dear Dr. Breiling,

Thank you very much for sending us the constructive and helpful referee's comments on our manuscript by Pfisterer et al. entitled "Insights from human NF- κ B knockouts". In the meantime, we have revised the manuscript, taking into consideration your editorial requests and the suggestions from the reviewers. All the changes in the manuscript are marked by writing in red font color.

As we reference code that was used for our analysis and figure generation, we have uploaded this code to zenodo. The code can be previewed under the following link and will be made public upon acceptance of the manuscript:

https://zenodo.org/records/15212566?preview=1&token=eyJhbGciOiJIUzUxMiJ9.eyJpZCI6ImZlODdjN2UyLTlkZGEtNGRhOS1hMmI4LWE5ZTM0N2FhZDk2OSIsImRhdGEiOnt9LCJyYW5kb20iOiIzZTUzYWZmNzRiYWVjMTMwZDdmODE0ZWQwZTIxMTIhMyJ9.xWo0uJtp7lz3ikbByCiM1dX5fKxplgSYnIMgf8dZtJVNVUA_oS1FeTtloezoqfEEvEuZdqFF9Fncr2z4EqqNHw

Below, please find the point-to-point answers to the questions, criticisms and queries written in blue:

Referee #1:

The manuscript by Pfisterer and colleagues leverages large genomic sequencing datasets to explore the essential or non-essential nature of genes encoding components of the NF- κ B pathway. This is a timely study given the available data from these genomic datasets and our understanding of individual components that control transcriptional responses to a diverse range of stimuli. Timely as well given the emerging realisation that mice are not human and the expanding list of genes for which there are substantial phenotypic differences between mice humans when mutated or deleted. The authors have presented a thorough, robust but very readable overview of human knockouts and the essential nature of NF- κ B components. By providing their analysis as supplementary files they also provide a resource to be utilised by others. The diagrams and figures are useful addition to what is a very interesting and thought provoking analysis. Given the relevance of NF- κ B to human disease and health this

paper will be high interest to a large number of researchers in these areas. I wholeheartedly recommend it for publication.

Minor comment.

It might be helpful to explain in a little more detail how gene essentiality was calculated (shet), perhaps in the "Box" section?

Answer: According the suggestion of the reviewer we added a short description of the key metrics to the "Caveats" box and refer to more detailed literature (Bartha *et al*, 2018; Rancati *et al*, 2018; Zeng *et al*, 2024).

Referee #2:

Pfisterer and colleagues hypothesise that non-essential genes might be good therapeutic targets because their inhibition is less likely to cause side effects. To identify such genes, they present an analysis aggregated datasets of human variants to estimate the burden of predicted LOF variants in genes relevant to NF- κ B in comparison to the remainder of the genome. The analysis is of interest but I have some reservations as outlined below:

1. In its current form, the manuscript is quite hard work. The supplementary tables are not labelled as such but appear as source files with no indication of which corresponds to each supplementary table in the text.

Answer: The reviewer is completely right, but upload of these data as supplementary tables was not possible in the PDF format. We have therefore uploaded the data as source files to preserve the Excel format and the important hyperlinks contained within. In the revised version of the manuscript, the source files are now correctly labeled as "Dataset EVx".

2. Nomenclature is at times ambiguous. For example, table 1 (and the relevant text) lists 6689 "non-redundant" genes. Two paragraphs later, the text refers to "strongly redundant genes", but these two uses of the term "(non)-redundant genes" are completely different. The former refers to the approach of compiling the gene list, and the latter refers to their biological function.

Answer: This point is very well taken, we therefore renamed sheet 1 of Dataset EV1 to "nonessential genes (studies)" and the full list of human nonessential genes to "list of nonessential genes".

3. The relation between gene names and protein names in the NF- κ B system is inherently confusing. The authors amplify this problem because throughout the text and figures they refer to genes using their protein names. The reader is left to reconcile the information in the text and figures with the supplementary tables that consist entirely of gene names.

[Answer: The defects in the genes ultimately lead to dysfunctional, absent, or severely truncated proteins. Since the entire NF- \$\kappa\$ B literature uses the protein names, we would like to maintain that convention. In Dataset EV2, we present both the gene names, the corresponding protein names, and the underlying literature. Each field is also linked to GeneCards[®] and PubMed entries, ensuring that all relevant information \(including alternative names\) is accessible. In the new Dataset EV3, we display the genes of the NF- \$\kappa\$ B system and highlight the non-essential genes in red. As shown in the screenshot, all gene names are hyperlinked so that the full information on the corresponding protein can be accessed immediately. Screenshots of both EV Datasets are provided here.](#)

E	F	G	H	I
Canonical NF-κB pathway				
protein	gene	PMID	pathway (L=LPS, I=IL1, T=TNF)	
TRADD	TRADD	8943045	TL	
FADD	FADD	https://www.genecards.org/cgi-bin/carddisp.pl?gene=TRADD&keywords=TRADD - Klicken Sie einmal, um dem Hyperlink zu folgen. Klicken Sie, und halten Sie die Maustaste gedrückt, um die Zelle auszuwählen.		
ciAP1	BIRC2	31209050		
ciAP2	BIRC3	31209050		
HOIL-1	RBCK1	20005846		
SHARPIN	SHARPIN	20005846	TL	

Protein names are indicated, all gene names and all references are hyperlinked.

	A	B	C	D	E
1	Gene				
2	AKT1				
3	ANXA5		Non-redundant list of genes (linked to the respective proteins via GeneCards) that participate in NF-κB activation in diverse pathways. Predicted knockouts are shown in red.		
4	ATM				
5	BCL10				
6	BCL3				
7	BIRC2				
8	BIRC3				
9	BLK				
10	BLNK				
11	BTk				
12	BTRC	https://www.genecards.org/cgi-bin/carddisp.pl?gene=BTk&keywords=BTk - Klicken Sie einmal, um dem Hyperlink zu folgen. Klicken Sie, und halten Sie die Maustaste gedrückt, um die Zelle auszuwählen.			
13	CAMK2				
14	CARD11				
15	CBLB				
16	CDK12				

4. On page 14, after describing each of the NF-κB pathways, the authors refer to the 28 knockout genes in the NF-κB system. It is possible that this list of 28 genes is somewhere in the tables but I couldn't locate it. Here and elsewhere, the authors could consider including summary tables within the main article that identify the genes of particular interest.

Answer: This is a good suggestion and also overlaps with Reviewer 3's "Other Comments". We therefore present all genes of the NF-κB system, with the non-essential genes highlighted in red, in the new Dataset EV3. The products of all these genes, as well as their positions within the activation pathways, are additionally shown in Figure 3.

5. There is some overlap between the datasets analysed (e.g. Regeneron and gnomAD). This will not affect the main findings of the analysis but will affect the interpretation of the extent of variants shared between them and should be mentioned.

Answer: This point is correct, and the information is also included in Table 1. As requested by the reviewer, we have now also addressed this fact in Box 1 (Caveats) with an additional sentence.

6. The variants mentioned in the discussion of the core NF-κB pathway are not all in the same category. The syndromes attributed to LoF of RELA and CHUK (Behcet's and fetal encasement/Bartsocas-Papas, respectively) are due to ultra-rare damaging variants, whereas psoriasis is only associated with polymorphisms in NFKBIZ. Furthermore, the analysis hinges on predictions of LOF from the aggregated datasets as indicators of their non-essentiality. While the authors acknowledge the limitations to this assumption, the analysis might be more balanced if it were to include data that has identified variants that are definitely pathogenic (e.g. ClinVar).

Answer: According to these suggestions we wrote: "Additionally, mutations in IκBζ have been associated with ulcerative colitis and psoriasis (Coto-Segura *et al*, 2017; Kakiuchi *et al*, 2020; Nanki *et al*, 2020), while mutations in IKKα lead to defects in the development of skin epidermis (Lahtela *et al*, 2010). However, these variants should be distinguished from the ultra-rare, highly damaging loss-of-function mutations observed in p65, further information on the clinical relevance of mutations in NF-κB knockouts are given in Dataset EV4." In this new file we give all the relevant ClinVar data for the human NF-κB knockouts.

7. The final section on PTMs seems like a separate study. The conclusion that genetic conservation might predict functional importance is hardly novel - it is after all the basis of almost all in silico predictors of damage conferred by missense mutations.

Answer: In response to this critique, we have thoroughly revised this section to present the chosen dual strategy more clearly and effectively. Our analysis is novel for two reasons and, to the best of our knowledge, has never been conducted in a comparable form.

I) We used the genome data of the sequenced individuals, which has only been available for

a few years, to analyze the mutation status of the post-translationally modified amino acids of the NF- κ B core proteins. Therefore, we first had to develop a software program that matches the modified amino acids listed in the PhosphoSitePlus database with the sequence variations contained in the gnomAD database. The corresponding program has also been uploaded to Zenodo (doi: 10.5281/zenodo.15212566) and is now available for use by the scientific community for similar analyses on any protein of interest. This analysis identified several PTM sites that were unaffected by missense mutations in the sequenced individuals.

II) These sites were also analyzed for their evolutionary conservation, as they are generally believed to have a higher probability to be functionally important (Beltrao *et al*, 2013).

Only through this combined analysis of intra-species conservation (deep sequencing of the human population) and inter-species conservation (analysis of evolutionary constraint) was it possible to generate a list of potentially relevant modification sites among the vast number of different PTM sites. As convincingly shown in Figure 4 for the example of I κ B α and I κ B β , this double filtering strategy can confirm the relevance of already known important modification sites. Therefore, we assume that this dual strategy can also be applied for the *in silico* prediction of functionally relevant PTM sites for any human protein.

8. The authors suggest that non-essential NF- κ B genes might be satisfactory therapeutic targets because their inhibition is predicted to cause fewer side effects. Another way of looking at it, however, is that inhibition of these genes might lack any therapeutic efficacy due to redundancy. The authors might comment on this possibility.

Answer: This critique is valid and overlaps with the critique from Referee #3, Point 1. We have therefore added the statement: Interestingly, also non-essential genes are interesting target structures, as an earlier study showed that approximately half of 383 FDA-approved drugs target the products of unconstrained genes (Minikel *et al*, 2020).

Referee #3:

Major Concerns:

First, The Genome Aggregation Database (GnomAD) Consortium published four articles in Nature in 2020. In one of these articles, by Minikel *et al* (doi.org/10.1038/s41586-020-2267-z), explained how human KO studies should interpret the LoF variants in drug development. Figure 1a in this article shows that the targets of 383 FDA-approved drugs contain both constrained and unconstrained genes in nearly similar proportions, as they exist in the total genome. The authors write in their summary: "Here we report three key findings regarding the assessment of candidate drug targets using human loss-of-function variants. First, even essential genes, in which loss-of-function variants are not tolerated, can be highly successful as targets of inhibitory drugs." This suggests that, as of now, there is no evidence that non-essentiality of a gene should be a criterion for a drug target. While it is possible that in the case of NF- κ B activity control non-essential factors will not be a better target, the authors must establish the logic coherently and with supporting evidence. I am totally surprised that Pfisterer *et al*. ignored this most relevant reference in their work. This is a significant oversight.

Answer: We have now cited this important review and adjusted the text accordingly to indicate that suitable drug targets are found in both constrained and unconstrained genes.

Second, I also do not understand the assertion that genes within the NF- κ B system are nonessential or redundant. The authors have not adequately established the criterion for

redundancy. All members (I κ B, NF- κ B or IKK) in a family within the NF- κ B system show partial redundancy.

Answer: Redundancy is briefly mentioned, as it is a fundamental characteristic of all biological systems, which serves to distribute functions across networks to confer robustness, buffering and resilience (Uda *et al*, 2013). In contrast, this review focuses exclusively on the issue of essentiality in the NF- κ B system, as this aspect has never been systematically addressed in a review before.

To establish essentiality is even more complex. Pfisterer *et al* write, "biallelic knockouts were predicted for p65", but I don't see evidence for this. Contrarily, Lek *et al* showed RelA has no LoF variants (Table 13).

Answer: Narasimhan *et al.* published a curated list of non-essential genes originally identified in the Lek study (Lek *et al*, 2016), with Monkol Lek included as a co-author (Narasimhan *et al*, 2016). This amended list was generated using additional criteria and is available as a supplementary dataset (File: aac8624_data_s2.xlsx), which includes the gene RELA. The study identified 1,775 knockout genes in the ExAC dataset, a number that is generally accepted in the field (Bartha *et al.*, 2018; Kruger, 2016; Rausell *et al*, 2020). Furthermore, in the gnomAD browser (Dataset v4.1.0), 21 individuals show a homozygous p.Asp347IlefsTer16 frameshift mutation in RELA, all of whom are of East Asian ancestry, based on genome and exome sequencing data.

In that list, Rel, RelB, NF- κ B1 and NF- κ B2 all show LoF variants, as do IKBKB (IKK2) and CHUK (IKK1), but not RelA. Moreover, Lek *et al.*, rarely discussed biallelic KO. Sun *et al* compiled a list of biallelic knockouts of 4848 genes and none of the NF- κ B family members, including RelA, appear in that list. Am I missing something here?

Answer: As shown in Figure 1A, the majority of knockouts are found in only one study, while 1,485 knockouts were found in two studies and 1,309 in ≥ 3 studies. All knockout genes can be downloaded from one of the supplementary tables of the respective studies summarized in Table 1, and a compilation can be found in Dataset EV1. As described in the manuscript, we do not consider monoallelic mutations but state: "Here we only consider mutations affecting both alleles of a gene as either homozygous or compound heterozygous mutations, where both alleles harbor different mutations."

Additionally, the authors write that "whether the 21 persons with a p65 knockdown (due to a biallelic stop downstream of Asp347) identified by the study by Lek and colleagues also suffer from inflammatory disease" (page 9, top). I haven't analyzed their data, but the idea of having 11 biallelic LoF variants for a single gene in only 60,000 individuals seems highly improbable.

Answer: I assume that the reviewer is referring to 21 knockouts instead of the 11 written? We state in the manuscript: "Approximately half of all mutations were observed as singletons, i.e. they occur only in one individual (Lek *et al.*, 2016; Sun *et al*, 2024)." The relative frequency of knockouts for a particular gene depends on the sequenced population. For example, in a bottlenecked population (such as those found in Iceland or northern Pakistan), an increased frequency of a mutation is to be expected compared to outbred populations. Due to this intrinsic bias, the relative knockout frequency is not considered here. Instead, we focus on the question of whether it is fundamentally possible for a defect in a specific gene to be compatible with development and viability.

The authors should check the data very carefully, considering that the 60,000 individuals are largely normal, not suffering from severe diseases. In contrast, inflammatory disorders reported by Adeeb *et al.* and Badran *et al.* are RelA heterozygous variants, which were found

due to the severity of these diseases. They should double check this information.

Answer: We have added the new Dataset EV4 which shows the clinical relevance of mutations in NF- \$\kappa\$ B knockouts from the ClinVar database. As summarized in Table 1, the health status of most sequenced individuals was not investigated.

Other comments:

The section on human knockouts in different NF- κ B activation pathways describes the pathways well but offers very little analysis of human knockouts. The 28 knockouts should be listed in a table, showing the specific mutations, whether they are truly biallelic or monoallelic, and other relevant details.

Answer: The data referenced by the reviewer are derived from the compiled list of human non-essential genes (Dataset EV1). The health status of most individuals included in the sequencing cohorts was not assessed. As suggested by the reviewer, we examined the homozygous predicted pLOF variants for the 28 genes within the NF- \$\kappa\$ B pathway and their respective ClinVar classifications using gnomAD data, as now provided in the new Dataset EV4.

The same critique applies to the discussion of post-translational modifications (PTMs): the analysis is quite superficial. There is nothing new except the observation that many critical, highly conserved modification sites do not appear as missense variants.

Answer: Suggestions for improvement for this section were also provided by Referee #2, point 8. We therefore refer to our detailed response above.

We thank the reviewer for the helpful comments which we believe have helped to clarify several important key aspects of the manuscript. All criticisms and key questions posed by referee could be fully addressed. We think that the manuscript has been significantly strengthened by the revision and hope that it can be further considered for publication. With best wishes,

Lienhard Schmitz

References

Bartha I, di Iulio J, Venter JC, Telenti A (2018) Human gene essentiality. *Nat Rev Genet* 19: 51-62

Beltrao P, Bork P, Krogan NJ, van Noort V (2013) Evolution and functional cross-talk of protein post-translational modifications. *Mol Syst Biol* 9: 714

Coto-Segura P, Gonzalez-Lara L, Gomez J, Eiris N, Batalla A, Gomez C, Requena S, Queiro R, Alonso B, Iglesias S *et al* (2017) NFKBIZ in Psoriasis: Assessing the association with gene polymorphisms and report of a new transcript variant. *Hum Immunol* 78: 435-440

Kakiuchi N, Yoshida K, Uchino M, Kihara T, Akaki K, Inoue Y, Kawada K, Nagayama S, Yokoyama A, Yamamoto S *et al* (2020) Frequent mutations that converge on the NFKBIZ pathway in ulcerative colitis. *Nature* 577: 260-265

- Kruger RP (2016) Knockout! Knockout! Who's Not There? *Cell* 167: 289-291
- Lahtela J, Nousiainen HO, Stefanovic V, Tallila J, Viskari H, Karikoski R, Gentile M, Saloranta C, Varilo T, Salonen R *et al* (2010) Mutant CHUK and severe fetal encasement malformation. *N Engl J Med* 363: 1631-1637
- Lek M, Karczewski KJ, Minikel EV, Samocha KE, Banks E, Fennell T, O'Donnell-Luria AH, Ware JS, Hill AJ, Cummings BB *et al* (2016) Analysis of protein-coding genetic variation in 60,706 humans. *Nature* 536: 285-291
- Minikel EV, Karczewski KJ, Martin HC, Cummings BB, Whiffin N, Rhodes D, Alfoldi J, Trembath RC, van Heel DA, Daly MJ *et al* (2020) Evaluating drug targets through human loss-of-function genetic variation. *Nature* 581: 459-464
- Nanki K, Fujii M, Shimokawa M, Matano M, Nishikori S, Date S, Takano A, Toshimitsu K, Ohta Y, Takahashi S *et al* (2020) Somatic inflammatory gene mutations in human ulcerative colitis epithelium. *Nature* 577: 254-259
- Narasimhan VM, Hunt KA, Mason D, Baker CL, Karczewski KJ, Barnes MR, Barnett AH, Bates C, Bellary S, Bockett NA *et al* (2016) Health and population effects of rare gene knockouts in adult humans with related parents. *Science* 352: 474-477
- Rancati G, Moffat J, Typas A, Pavelka N (2018) Emerging and evolving concepts in gene essentiality. *Nat Rev Genet* 19: 34-49
- Rausell A, Luo Y, Lopez M, Seeleuthner Y, Rapaport F, Favier A, Stenson PD, Cooper DN, Patin E, Casanova JL *et al* (2020) Common homozygosity for predicted loss-of-function variants reveals both redundant and advantageous effects of dispensable human genes. *Proc Natl Acad Sci U S A* 117: 13626-13636
- Sun KY, Bai X, Chen S, Bao S, Zhang C, Kapoor M, Backman J, Joseph T, Maxwell E, Mitra G *et al* (2024) A deep catalogue of protein-coding variation in 983,578 individuals. *Nature* 631: 583-592
- Uda S, Saito TH, Kudo T, Kokaji T, Tsuchiya T, Kubota H, Komori Y, Ozaki Y, Kuroda S (2013) Robustness and compensation of information transmission of signaling pathways. *Science* 341: 558-561
- Zeng T, Spence JP, Mostafavi H, Pritchard JK (2024) Bayesian estimation of gene constraint from an evolutionary model with gene features. *Nat Genet* 56: 1632-1643

Referee #2:

I thank the authors for their careful and thoughtful revisions of their manuscript. In particular, I am pleased to see the updated tables.

I have a remaining concern about the classification of non-essential genes. The authors define essential genes as those that are 'indispensable for reproductive success and maintenance of a normal phenotype'. I take this definition to mean that genes for which biallelic or homozygous LoF variants confer a disease phenotype are essential. If this is correct, then I have three comments:

1. Tables EV3/4 and Fig 3 summarise non-essential NF- κ B genes. Following on from my recommendation, table EV4 includes information from ClinVar. As would be expected, there is limited overlap between variants in ClinVar and gnomAD variants (<5% of the 9.3×10^6 missense SNVs in gnomAD are in ClinVar). In any case, for 3 of 28 genes, LoF gnomAD variants are listed as pathogenic or likely pathogenic. For example, ATM, a gene actually named for the pathology resulting from its pathogenic variants, has one homozygous LoF entry in gnomAD. What makes this gene non-essential?

2. In discussing RELA (p65) the authors make the following statement regarding gnomAD LoF variants.

'However, these variants should be distinguished from the ultra-rare, highly damaging loss-of-function mutations observed in p65, further information on the clinical relevance of mutations in NF- κ B knockouts are given in dataset EV4.'

Of course, missense mutations may vary in their effects, but I don't understand how LoF variants can vary in severity. The assumption is that nonsense and canonical splice site variants results in complete loss of protein. If there is a distinction I am missing, why would the phenotype of a true LoF mutation not count? Furthermore, any LoF variant in an essential gene is expected to be ultrarare (or even absent) because of purifying selection. Could the authors clarify this basis of the distinction mentioned in quotation marks above?

3. If genes are included on the non-essential list despite instances of LoF variants that are pathogenic, is this because of an assumption of incomplete penetrance? If so, have the authors decided on a penetrance threshold, how has this been calculated, and based on what data?

Referee #3:

While the manuscript has been substantially improved after revision, I still have some concerns regarding the interpretation of certain gene knockouts, particularly how the term "knockout" is being defined in the context of genome sequencing data.

The characterization of RelA (which should be referred to as p65/RelA for clarity) as a knockout based on a premature stop codon at position 347 may be misleading. This truncation likely does not fully eliminate function. First, the remaining portion of RelA may still heterodimerize with other NF- κ B subunits such as c-Rel or RelB. Second, even in its truncated form, the DNA-binding domain might retain some ability to activate transcription,

potentially through mechanisms independent of direct DNA binding. Therefore, categorizing RelA as non-essential based solely on this variant warrants more nuanced interpretation. The question remains: given this partial functionality, is it truly accurate to classify RelA as non-essential?

Similarly, the distinction made between CHUK (IKK α /IKK1) and IKBKB (IKK β /IKK2) in terms of essentiality could be reconsidered. While CHUK is referred to as non-essential, there are several reported missense mutations in IKBKB (e.g., PMID: 33658989, 35067643, 25139357) that suggest this gene may also tolerate certain functional alterations. These variants may not be observed in large-scale datasets such as those derived from one million whole-genome sequences, but their documented existence should temper the assertion of IKBKB's essentiality.

Overall, I recommend that the authors clarify their criteria for defining "knockout" and "non-essential" within the context of population genomics data, and consider the functional nuances when making these classifications.

Introductory remark: Both reviewers have further questions and doubts regarding the validity of the knockout of individual genes, which I will address in detail below. However, I would first like to reiterate that the human knockout genes were not identified and defined by the authors of this review, but rather in various studies that are also cited in the review. These nine studies are summarized in Table 1 and have been published in high-ranking journals, namely:

Science (MacArthur *et al*, 2012; Narasimhan *et al*, 2016a)

Nat. Genet. (Sulem *et al*, 2015)

Nature (Karczewski *et al*, 2020; Lek *et al*, 2016; Saleheen *et al*, 2017; Sun *et al*, 2024)

PLoS Genet (Lim *et al*, 2014) and

Proc Natl Acad Sci U S A (Rausell *et al*, 2020)

The authors of these studies are highly respected scientists, and articles in the aforementioned journals undergo rigorous and stringent peer review processes. Therefore, we are not in a position to seriously question the validity of the identification of these respective knockout genes. We thus base this review on the published results. Of course, the limitations described in Box 1 "Caveats" apply to all of these studies.

Below, please find the point-to-point answers to the questions, criticisms and queries written in blue:

Referee #2:

1) I have a remaining concern about the classification of non-essential genes. The authors define essential genes as those that are 'indispensable for reproductive success and maintenance of a normal phenotype'. I take this definition to mean that genes for which biallelic or homozygous LoF variants confer a disease phenotype are essential. If this is correct, then I have three comments:

Answer: This assumption is not entirely correct, as we wrote on page 6: "Additional information about the health status of the sequenced individuals is highly variable. In many studies, the health status of the analyzed individuals was not recorded,..." The details regarding the recording of the participants' health status can also be found in the right-hand column of Table 1.

To emphasize the largely missing information on the health status even more, we added the following statement to the revised manuscript: "In many studies, the health status of the individuals examined was not recorded, and potential health problems may appear only later in life."

1. Tables EV3/4 and Fig 3 summarise non-essential NF-kB genes. Following on from my recommendation, table EV4 includes information from ClinVar. As would be expected, there is limited overlap between variants in ClinVar and gnomAD variants (<5% of the 9.3×10^6 missense SNVs in gnomAD are in ClinVar). In any case, for 3 of 28 genes, LoF gnomAD variants are listed as pathogenic or likely pathogenic. For example, ATM, a gene actually named for the pathology resulting from its pathogenic variants, has one homozygous LoF entry in gnomAD. What makes this gene non-essential?

In the case of ATM, we can only speculate and refer to the various possibilities listed in Box 1 "Caveats." We mention there, among other things, that the essentiality of genes is determined also by the respective epigenetic or genetic background. Furthermore we do not

know the health status of most of the sequenced individuals, so issues similar to those observed in adult ATM knockout mice (Elson *et al*, 1996) cannot be ruled out.

2. In discussing RELA (p65) the authors make the following statement regarding gnomAD LoF variants.

'However, these variants should be distinguished from the ultra-rare, highly damaging loss-of-function mutations observed in p65, further information on the clinical relevance of mutations in NF- κ B knockouts are given in dataset EV4.'

Of course, missense mutations may vary in their effects, but I don't understand how LoF variants can vary in severity. The assumption is that nonsense and canonical splice site variants results in complete loss of protein. If there is a distinction I am missing, why would the phenotype of a true LoF mutation not count? Furthermore, any LoF variant in an essential gene is expected to be ultrarare (or even absent) because of purifying selection. Could the authors clarify this basis of the distinction mentioned in quotation marks above? To clarify this issue we deleted the sentence about the highly damaging loss-of-function mutations observed in p65 and added the following statement that clarifies the questions of the reviewer:

How is it possible that knockout carriers of these genes are presumably symptom-free, while mutations in these genes are associated with inflammatory diseases?

(I) The type of genetic alteration has a decisive impact on the resulting phenotype. A well-studied example is the knockout of the gene encoding the cellular prion protein (PrP), which shows no obvious phenotype in mice (Büeler *et al*, 1992). In contrast, mice with a point mutation in the PrP protein show spontaneous PrP aggregation and develop Spongiform encephalopathy (Sigurdson *et al*, 2011). The functional difference between the complete loss of a protein and its mutated form is also relevant to human disease. An example of this is the p53-coding gene, which is rarely deleted in tumor patients but in the vast majority of cases exhibits oncogenic p53 missense mutations that enable the gain of new harmful functions, thereby promoting cancer development (Lane, 2024). Many further studies observed incomplete matching between human gene knockouts and disease-causing gene mutations. For example, it is well known that deafness can be caused by *GJB2* gene mutations (omim.org, (Wilcox *et al*, 2000)), while individuals with a *GJB2* knockout exhibit normal audiometry (Narasimhan *et al.*, 2016a). For further examples describing lacking congruence between disease-causing mutations and human knockouts we refer to the relevant literature (Narasimhan *et al*, 2016b).

(II) Another explanation is derived from the observation that the impact of a specific gene loss also depends on its interaction with the genome. The penetrance of a given mutation is often dependent on the genetic context, in which so-called modifier genes, which do not cause disease on their own, can influence the effect of a disease-causing mutation. A well-studied example are cystic fibrosis transmembrane conductance regulator (CFTR) gene mutations, which can cause cystic fibrosis (CF) depending on the levels of transforming growth factor (TGF)- β 1 (Najm *et al*, 2024). In addition, such differences can be attributed to differences in the types of mutations.

3. If genes are included on the non-essential list despite instances of LoF variants that are pathogenic, is this because of an assumption of incomplete penetrance? If so, have the authors decided on a penetrance threshold, how has this been calculated, and based on

what data?

This question overlaps with question 2 and we refer to the answer given above. There we argue in detail why knockout carriers of specific genes are presumably symptom-free, while mutations in these genes can be associated with diseases. As described in the manuscript and re-instated in the introductory remark, we did not define the knockouts and the mathematical thresholds, but rather the authors of the studies summarized in Table 1.

Referee #3:

1. The characterization of RelA (which should be referred to as p65/RelA for clarity) as a knockout based on a premature stop codon at position 347 may be misleading. This truncation likely does not fully eliminate function. First, the remaining portion of RelA may still heterodimerize with other NF- κ B subunits such as c-Rel or RelB. Second, even in its truncated form, the DNA-binding domain might retain some ability to activate transcription, potentially through mechanisms independent of direct DNA binding. Therefore, categorizing RelA as non-essential based solely on this variant warrants more nuanced interpretation. The question remains: given this partial functionality, is it truly accurate to classify RelA as non-essential?

To answer this question, two aspects are important. First, it was not we as the authors of this study who defined the criteria for knockouts, but rather the authors of the sequencing studies listed in Table 1. As stated in Box 1 "Caveats", a mutation does not always result in a complete loss of function, and the essentiality of genes is determined by the respective epigenetic or genetic background. Therefore, we cannot exclude residual functions for the knockout genes listed here and explicitly point out this fact.

Second, in the case of p65, the significantly truncated p65 protein is no longer capable of gene activation, as we have published (Schmitz & Baeuerle, 1991). We also found that a p65 mutant that cannot bind DNA loses all of its transcriptional and biological functions (Riedlinger *et al*, 2019).

2. Similarly, the distinction made between CHUK (IKK α /IKK1) and IKBKB (IKK β /IKK2) in terms of essentiality could be reconsidered. While CHUK is referred to as non-essential, there are several reported missense mutations in IKBKB (e.g., PMID: 33658989, 35067643, 25139357) that suggest this gene may also tolerate certain functional alterations. These variants may not be observed in large-scale datasets such as those derived from one million whole-genome sequences, but their documented existence should temper the assertion of IKBKB's essentiality.

As already stated in the response to Referee 1, point 2, it is important to clearly distinguish between two things here. The knockout of a gene may result in a different phenotype compared to point mutations of the same gene. We illustrate this fundamental genetic principle using the examples of PrP and p53 proteins and the *GJB2* gene.

3. Overall, I recommend that the authors clarify their criteria for defining "knockout" and "non-essential" within the context of population genomics data, and consider the functional nuances when making these classifications.

In the manuscript we clearly describe these criteria:

Knockout = Identified in one of the nine studies mentioned above. These genes may be completely or at least largely inactivated, making it a **physical property**.

At the same time, the knockout genes are non-essential, meaning they are not necessary for the development and survival of the organism, which represents a **functional property**.

Prof. M. Lienhard Schmitz
Justus-Liebig University
Institute for Biochemistry
Friedrichstrasse 24
Giessen 35392
Germany

Dear Prof. Schmitz,

Thank you for the submission of your final revised manuscript to our editorial offices. I have now received the report from one of the two referees that were asked to look into it, you will find below. As you will see, the referee now supports the publication of your review in EMBO reports. Original referee #2 was unresponsive to my invitation to re-assess the manuscript. However, going through your last p-b-p-response, I consider his/her remaining points as adequately addressed. Referee #1 already supported publication of the initial version.

I am thus pleased to inform you that your manuscript has been accepted for publication in EMBO reports. Your manuscript will be processed for publication by EMBO Press. It will be copy edited and you will receive page proofs prior to publication.

Please make sure that the externally deposited code is public latest upon online publication of the manuscript.

You will soon be contacted by Springer Nature to sign your publishing license. When you login to the customer service website, please use the token/code copied below to waive the article publication charges. Should you experience any difficulty, please email publishing@embo.org.

Token/code: LTY4MJGXMJK1MA

If you have any other questions, please do not hesitate to contact the Editorial Office. Thank you for your contribution to EMBO Reports.

Yours sincerely,

Referee #3:

After careful consideration, I accept this further revised manuscript. To the best of my knowledge, this is the first manuscript of its kind on NF- κ B signaling. This review will encourage researchers to think deeply about gene knockouts in humans and their essentiality. With further work on this subject, many questions currently on my mind may be clarified. Overall, this is a solid piece of work, and I support its publication.
